



# Biogeochemical evidence of anaerobic methane oxidation and anaerobic ammonium oxidation in a stratified lake using stable isotopes

Florian Einsiedl[1], Anja Wunderlich[1], Mathieu Sebilo[2], Ömer K. Coskun[3], William D. Orsi[3,4], Bernhard Mayer[5]

[1]Chair of Hydrogeology, Technical University of Munich, TUM Department of Civil, Geo and Environmental Engineering, Arcisstrasse 21, 80333 München, Germany

[2]Sorbonne Université, CNRS, IEES, F-75005, Paris, France

[3]Department of Earth and Environmental Sciences, Paleontology & Geobiology, Ludwig-Maximilians-Universität München, 80333 Munich, Germany

[4]GeoBio-Center, Ludwig-Maximilians-Universität München, 80333 Munich, Germany

[5]Department of Geoscience, University of Calgary, Calgary, Alberta, Canada T2N 1N4

Correspondence to: Florian Einsiedl (f.einsiedl@tum.de)

**Abstract.** Nitrate pollution of freshwaters and methane emissions into the atmosphere are crucial factors in deteriorating the quality of drinking water and in contributing to global climate change. The nitrate dependent anaerobic methane oxidation and the anaerobic oxidation of ammonium (*anammox*) have the potential to reduce nitrogen loading of aquatic ecosystems and to reduce methane emissions to the atmosphere.

Here, we report vertical concentration profiles and corresponding stable isotope compositions of $CH_4$, $NO_3^-$, $NO_2^-$ and $NH_4^+$ in the water column of a stratified lake, which suggest linkages between anaerobic oxidation of methane (AOM), denitrification, and *anammox*. In a water depth from 12 to 20 m, a methane-nitrate transition zone (NMTZ) was observed, where $\delta^{13}C$ values of methane and $\delta^{15}N$ and $\delta^{18}O$ of dissolved nitrate markedly increased in concert with decreasing concentrations of methane and nitrate. These data patterns, together with a simple 1D diffusion model show that the non-linear methane concentration profile cannot be explained by diffusion or micro-aerobic methane oxidation, and that microbial oxidation of methane coupled with denitrification under anaerobic conditions is the most likely explanation for these data trends.

In the methane zone at the bottom of the NMTZ (20 m to 22 m) $\delta^{15}N$ of ammonium increased by 4‰, while ammonium concentrations decreased. In addition, a strong $^{15}N$ enrichment of dissolved nitrate was observed at a water depth of 20 m, suggesting that *anammox* is occuring together with denitrification coupled to AOM. The conversion of nitrite to $N_2$ and nitrate during *anammox* is namely associated with an inverse N isotope fractionation and may explain the observed increasing offset $(\Delta\delta^{15}N)$ of 26 ‰ between $\delta^{15}N$ values of dissolved nitrate and nitrite at a water depth of 20 m compared to the $\Delta\delta^{15}N_{nitrate-nitrite}$ of 11 ‰ obtained in the NMTZ between a water depth of 16 m and 18 m.



The geochemcical zones were found to contain significantly different microbial communities that consist of bacteria known to be involved in denitrification with AOM (*Crenothrix* and NC10), and *anammox* ('*Candidatus* Anammoximicrobium'), confirming the presence of microbial groups potentially responsible for the proposed linkages between AOM, denitrification, and *anammox*. This study gives insights into the yet overlooked AOM-denitrification-anammox process in stratified lakes that can regulate methane emisssions from and nitrogen concentrations in lakes.

## 1    Introduction


Methane is a more potent greenhouse gas than $CO_2$ and is responsible for 20% of global warming (Change, 2001). Bastviken et al. (2004) have shown that lacustrine ecosystems may be responsible for 6 to 16 % of natural methane emission. However, the variability in methane emissions and the lack of knowledge about their main environmental controls contribute large uncertainties into the global $CH_4$ budget (Sabrekov et al., 2017).

Methane is abundantly formed in anaerobic lake sediments by methanogenesis (Borrel et al., 2011; Conrad et al., 2007; Norði et al., 2013) and diffuses upwards through the water column toward the oxycline of often nitrate-containing seasonally stratified lakes. With the discovery of the anaerobic oxidation of methane (AOM) couples to nitrate or nitrite reduction more than 10 years ago a new process was suggested that has the potential to reduce emissions of greenhouse gases of lacustrine environments by converting $CH_4$ to more harmless and soluble $CO_2$ (Ettwig et al., 2010; Haroon et al., 2013; Raghoebarsing

et al., 2006). It was experimentally shown that *n-damo* (nitrite depending anaerobic oxidation of methane) bacteria that are members of the candidate phylum NC10 use nitrite for the anaerobic oxidation of methane (Ettwig et al., 2010), while archaea such as ANME-2d prefer nitrate as electron acceptor (Haroon et al., 2013). Evidence of archaeal AOM coupled with bacterial denitrification was first reported from culture experiments with two microorganisms, '*Candidatus* Methylomirabilis oxyfera*'* which belongs to the phylum NC10 and reduces nitrite to $N_2$, whereas ANME-2d lineage  affects the reduction of nitrate to

nitrite (Raghoebarsing et al., 2006).

 Microbial nitrate reduction may be more relevant in aquatic environments than nitrite reduction due to the significantly higher concentrations of nitrate compared to nitrite in lake water or groundwater systems. Beside archaea like ANME-2d, more recently it was experimentally demonstrated and supported by genome analysis that filamentous gamma-proteobacterial methane oxidizing bacteria related to *Crenothrix* have also the potential for methane-dependent growth under nitrate-reducing

conditions and under aerobic conditions (Kits et al., 2015; Naqvi et al., 2018; Oswald et al., 2017). Therefore, *Crenotrix* may likely act as a driver for methane oxidation in mainly nitrate-containing stratified lakes, where environmental and redox conditions can often change over seasonal periods.

A few environmental studies have documented the presence of NC10 like bacteria in lake sediments, which are thought to have a similar metabolism to '*Ca*. M. oxyfera'. Via micro-sensor measurements and molecular biological analysis it was





postulated that '*Ca. M. oxyfera*' is responsible for *n-damo* in the sediments of Lake Constance (Deutzmann et al., 2014), while others found some evidence of *n-damo* in the sediments of a lake in Japan (Kojima et al., 2012).

It has been speculated that denitrification can co-occur with *anammox* at oxic-anoxic interfaces (Strous and Jetten, 2004; Thauer and Shima, 2008). In the late 1980s, microorganisms driving the *anammo*x reaction were first discovered in a wastewater pilot plant (Francis et al., 2007; Mulder et al., 1995). Subsequently, the significance of the *anammox* process in the

nitrogen cycle of freshwater systems was shown in numerous studies (e.g. Schubert et al., 2006) and it was suggested that the process is of key environmental significance (Kuypers et al., 2003). The coexistence of denitrification coupled with AOM together with *n-damo* and *anammox* was clearly demonstrated in bioreactor studies supplied with nitrate, methane and ammonium (Haroon et al., 2013; Hu et al., 2015; Luesken et al., 2011; Shi et al., 2013). In addition, a limited number of studies exists for aquatic environments that demonstrate the co-occurrence of *n-damo* and *anammox (Shen et al., 2014; Zhu et al.,*

*2018)*. To obtain an accurate estimation of methane fluxes to the atmosphere and to identify the factors driving and limiting the reduction of nitrate and its intermediates in lacustrine environments we, therefore, have to improve our understanding of the link between the carbon and nitrogen cycles in lakes.

Stable isotope fractionation has often been used to identify microbial transformation processes affecting nitrogen and carbon including denitrification and AOM (e.g. Wunderlich et al., 2012). Recently, Granger and Wankel (2016) showed that

displaying the isotope compositions of nitrate in a 2D isotope plot ($\delta^{18}O$/ $\delta^{15}N$) enables the distinction between denitrification and *anammox*. In addition, aerobic and anaerobic methane oxidation was often documented by increasing $\delta^{13}C$ values in the remaining methane (Eller et al., 2005; Feisthauer et al., 2011). However, the separation of aerobic and anaerobic oxidation of methane based on calculated isotope enrichment factors of methane may fall short because of overlapping isotope values (Feisthauer et al., 2011).

Here we report chemical and isotopic evidence together with quantitative gene sequencing, and high-throughput Illumina sequencing of 16S rRNA genes data that provide effidence for the presence of denitrification linked with AOM and the occurrence of *anammox* in a freshwater habitat. To support our interpretation of the observed vertical concentration profile of methane within the water column of the stratified Lake Fohnsee, we applied a simple 1D-diffusion model and coupled the diffusion model with a degradation term to clarify the effect of dissolved oxygen on methane oxidation. The observed coupled

process has the potential to constitute an important sink of dissolved nitrogen ($NO_3^-$, $NO_2^-$, $NH_4^+$) and methane ($CH_4$) in freshwater environments.

## 2  Material and Methods
## 2.1  Field Site

The Ostersee lakes are located in Southern Germany and consist of a series of lakes that are hydrologically connected (Braig et al., 2010). The chain of lakes was formed after the rapid disintegration of the last ice sheet at the end of the Pleistocene. Lake Fohnsee which was sampled in 2016 is one of the Ostersee lakes. The lake is circa 22 m deep, fed by groundwater and





is stratified during summer with an oxic zone (epilimnion) near the surface and an anoxic redox zone (hypolimnion) below a water depth of approximately 12 m.

## 2.2 Sampling

A field campaign at lake Fohnsee was performed in summer 2016 to obtain depth-resolved water samples throughout the water column of the lake to a depth of 22 m. During the field campaign a submersible probe with sensors for temperature and oxygen content was used. Dissolved oxygen concentrations and lake water temperatures with a depth resolution of 1 m were measured on site. Water samples were taken with a discrete 2 L sampling unit ("Ruttner bottle") with a depth-resolution between 1 and 2 m. The detection limit of the oxygen-sensor FDO 925, WTW, Xylem, Germany) was < 0.625 µmol/L ($\leq$ 0.02 mg/L), the analytical error was 0.5 % of the measured value for oxygen. In addition to the in-situ measurements, samples for the laboratory-based measurement of major anion and cation concentrations, and water isotopes ($\delta^2$H, $\delta^{18}$O) were field-filtered with 0.2µm PES filters and stored in airtight 1.5 ml glass vials. Samples for isotope analysis of nitrite ($\delta^{15}$N, $\delta^{18}$O), nitrate ($\delta^{15}$N, $\delta^{18}$O) and ammonium ($\delta^{15}$N) were field-filtered with 0.2µm PES filters and stored in PE vials. Isotope samples were frozen at -23°C until processing. Samples for analysis of DOC (Dissolved Organic Carbon) concentrations were collected in 50 ml glass bottles, filtered with 0.45 µm PVDF filters and measured immediately in the laboratory. Samples for the concentrations and isotope analysis of methane ($\delta^{13}$C) were transferred into 200 ml glass vials without headspace and sealed with crimped butyl stoppers. Samples for molecular-biological investigations were collected in 2 L sterile glass bottles. Subsequently the 2L water samples were divided in two 1L samples for replicate measurements and each sample was filtered in the laboratory using a 0.2 µm sterile filter and kept frozen at -23°C prior to analysis.

## 2.3 Determination of water chemistry and DOC

The samples were analyzed with ion chromatography for concentrations of nitrate, nitrite, ammonium, and sulfate. The analyses were performed in triplicate using two parallel Thermo Scientific ICS1100 instruments with CS12A (cations) and AS9-HC (anions) columns, respectively. Values are reported as mean values (n=3) with an uncertainty of less than 10%. The detection limits for nitrate, nitrite, and ammonium are < 0.15 mg/L.

DOC concentrations were determined by lowering the pH of the samples to remove inorganic carbon and subsequent spectral analysis of $CO_2$ after combustion (Analytic Jena Multi N/C 3100) with a measurement uncertainty of ±5% and a detection limit of 0.5 mg/L.

## 2.4 Analytical model to evaluate methane diffusion and the potential of micro-aerobic oxidation of methane in the water column





For the 1-D diffusion model, a semi-infinite system was assumed where x = 0, the boundary is kept at a constant input concentration $C_0$, and the initial concentration throughout the system is zero. The following formula (Eq. 1) from Crank (1975) was used to determine the methane concentration in 0.1 m intervals along the 10 m long water column below the oxycline at
time t:

$$C = C_0 \, erfc \; x = \frac{x}{\sqrt{2(Dt)}} \qquad\qquad\qquad \text{(Eq. 1)}$$

where $C_0$ [mg/L] is the initial concentration, C [mg/L] is the concentration at a distance x [m] at time [t] from the boundary at
a water depth of 22 m for methane and 12 m for oxygen, D [m² day$^{-1}$] is the turbulent diffusion coefficient for methane and the diffusion coefficient for oxygen in water and t [days] is the time it takes for methane and oxygen to diffuse a given distance in the water column where no mixing has occurred (90 days). For methane two turbulent diffusion coefficients of 0.1 and 2.1 m$^2$ day$^{-1}$, which have been typically applied for methane flux calculations and modeling at low turbulence levels at Lake Rotsee and Lake Lugano, respectively, were applied for the model runs (Oswald et al., 2015; Wenk et al., 2014). For dissolved oxygen
a diffusion coefficient of 0.2 m$^2$ day$^{-1}$ was used (Li, 1973). Modelling was performed downwards from 12 m water depth to the bottom at 22 m for dissolved oxygen and upwards from bottom at a water depth of 22 m to a water depth of 12 m for methane.

In order to add microbial degradation to the analytical model, the aerobic oxidation potential of methane was considered using equation 2 from Crank (1975) in a semi-infinite system with a constant concentration flux of oxygen at depth.
The equation 2 (Eq. 2) which was used for the 1-D diffusion and degradation model is given as (Crank, 1975):

$$C = C_0 \exp \frac{-x\sqrt{k}}{\sqrt{D}} \qquad\qquad\qquad \text{(Eq. 2)}$$

where k is the first-order oxidation rate constant [d$^{-1}$]. Here we used a value of 0.07 d$^{-1}$ which may represent the lower bound
of literature data (Blees et al., 2014; Zimmermann et al., 2019).

## 2.5 Measurement of stable isotope ratios

The natural abundance stable isotope ratios of nitrogen ($^{15}$N/$^{14}$N) in $NH_4^+$, $NO_3^-$, $NO_2^-$ and oxygen ($^{18}$O/$^{16}$O) in $NO_3^-$ and $NO_2^-$ as well as carbon isotope ratios ($^{13}$C/$^{12}$C) of methane constitute a powerful tool to identify biogeochemical transformation
processes involving these compounds. During AOM and denitrification the lighter isotopes ($^{12}$C, $^{14}$N, $^{16}$O) react preferentially leading to an enrichment of the heavier isotopes ($^{13}$C, $^{15}$N, $^{18}$O) in the residual substrate pool ($CH_4$, $NO_3^-$, $NH_4^+$) and an enrichment of the lighter isotopes in the newly formed products $CO_2$, $NO_2^-$, and $N_2$. Stable isotope ratios of C, N and O are





reported using the conventional delta (δ) notation expressed as $\boldsymbol{\delta = \left( \frac{R_{sample}}{R_{standard}} - 1 \right)}$ [‰] where $R_{sample}$ and $R_{standard}$ are the ratios of heavy versus light isotopes in the sample and an international standard, respectively.


### 2.5.1 Water isotope composition

Hydrogen and oxygen isotope ratios of water ($^{18}O/^{16}O$ and $^2H/^1H$) were analyzed by off-axis laser spectroscopy using a water analyzer (Los Gatos Instruments IWA-45EP) with a precision of 0.1‰ for $\delta^{18}O$ and 0.5‰ for $\delta^2H$ and are reported with respect to Vienna Standard Mean Ocean Water (V-SMOW).

### 2.5.2 Isotope compositions of nitrate, nitrite and ammonium

$\delta^{15}N$ and $\delta^{18}O$ values of nitrate and nitrite as well as $\delta^{15}N$ values of ammonium were obtained by the production of $N_2O$ following modified protocols of procedures reported by McIlvin and Altabet (2005), Semaoune et al. (2012) and Zhang et al. (2007). Nitrite was converted to $N_2O$ using sodium azide. In a second aliquot of the sample, nitrate was first reduced to nitrite in an activated column of cadmium and the mixture of both nitrate and nitrite was reduced to $N_2O$. The yield of conversion 170 was better than 95%. Nitrogen and oxygen isotope ratios of nitrate were calculated by measuring nitrite alone as well as the mixture of nitrite and nitrate in a sample and using an inverse mixing calculation to determine the isotopic ratios of nitrate alone. Ammonium was oxidized to nitrite using hypobromite ($BrO^-$). The nitrite produced from ammonium oxidation was then transformed into dissolved $N_2O$ by an azide buffer for subsequent analysis. The isotope compositions of all $N_2O$ samples were measured with an isotope ratio mass spectrometer (IRMS, Delta Vplus, Thermo Scientific, Bremen, Germany) in continuous-175 flow mode with a purge-and-trap system coupled with a Finnigan GasBench II system (Thermo Scientific, Bremen, Germany). Results are reported in the internationally accepted delta notation in ‰ with respect to the standards AIR for $\delta^{15}N$ and Vienna Standard Mean Ocean Water (V-SMOW) for $\delta^{18}O$. Ammonium, nitrate and nitrite reference materials subject to the same analytical procedures were used to calibrate the isotopic composition of $N_2O$. The standards USGS25, $\delta^{15}N$ = -30.4‰, IAEA-N1, $\delta^{15}N$ = 0.4‰, IAEA-N2, $\delta^{15}N$ = 20.3‰, IAEA-305, $\delta^{15}N$ = 39.8‰ were used for ammonium reference materials and 180 USGS34, $\delta^{15}N$ = -1.8‰, $\delta^{18}O$ = -27.9‰, USGS35, $\delta^{15}N$ = +2.7‰, $\delta^{18}O$ = +57.5‰ and USGS32, $\delta^{15}N$ = +180‰, $\delta^{18}O$ = +25.7‰ were used to calibrate nitrate measurements; Laboratory nitrite standards Lb1, $\delta^{15}N$ = -63‰ and Lb2, $\delta^{15}N$ = +2.7‰ were used to calibrate nitrite isotope analyses). The precision for $\delta^{15}N$ values of ammonium was ± 0.3‰. The precision for $\delta^{15}N$ values was ± 0.5‰ and for $\delta^{18}O$ ± 0.8‰ of nitrite and nitrate.

### 2.5.3 Concentrations and carbon isotope ratios of dissolved methane





The concentrations and carbon isotope ratios of dissolved methane in the lake water samples were determined using the static headspace equilibrium technique(EPA, 2002) where 10% of the water sample in the capped bottles was replaced with helium followed by outgassing of the dissolved gases in the water sample into the headspace for 1 h at 25ºC. Subsequently, the concentration of methane in the headspace was determined using gas chromatography (Bruker 450) with a measurement

uncertainty of $< \pm 5\%$. The concentration of dissolved methane in the water samples (in mg/L) was subsequently determined using Henrys Law (EPA, 2002).

The carbon isotope ratios of methane in the headspace samples were analyzed on a ThermoFisher MAT 253 isotope ratio mass spectrometer (IRMS) coupled to Trace GC Ultra + GC Isolink (ThermoFisher). Carbon isotope ratios of methane are reported in the standard delta notation in ‰ relative to the VPDB standard. The precision for carbon isotope analyses on dissolved

methane was better than $\pm 0.5$‰.

### 2.6 DNA extraction

Microbial biomass was collected on 0.22 µm cellulose acetate filters (Corning Inc., 1 NY, USA) in the laboratory after sampling and stored frozen on dry ice and later at -23°C until DNA extraction. Total DNA for groundwater microbial community analysis was extracted from frozen filters as previously described (Brielmann et al., 2009).

### 2.7 Quantitative gene sequencing

Quantitative PCR (qPCR) was performed using the custom primer dual indexed approach that is commonly applied in microbial ecology community analyses (Kozich et al., 2013), and targets the V4 hypervariable region of the 16S rRNA gene using updated 16S rRNA gene primers 515F/806R (515F: 5′ – GTGYCAGCMGCCGCGGTAA– 3′, 806R: GGACTACNVGGGTWTCTAAT) as described previously (Coskun et al., 2018). These 'universal' primers cover all major

groups of Bacteria and Archaea, and have the 'Y' ambiguity code insertion into the 515F forward primer to increase the coverage of Archaea (Parada et al., 2016). qPCR reactions were prepared using an automated liquid handler (pipetting robot), the EpMotion 5070 (Eppendorf), was used to set up all qPCR reactions and standard curves. The efficiency values of the qPCR were <90% and $R^2$ values >0.95% qPCR was performed using white 96-well plates. The technical variability of 16S rRNA gene qPCR measurements was determined to be consistently <5% under the EpMotion 5070.

Barcoded V4 hypervariable regions of the amplified 16S rRNA genes from the qPCR were sequenced on an Illumina MiniSeq following an established protocol (Pichler et al., 2018). This yielded a total of >2 000 000 raw sequencing reads that were then subjected to quality control. In order to quality control the OTU picking algorithm for the data, we also sequenced a "mock community" alongside our environmental samples. The mock communities contained a defined number of species (n=18) all containing 16S rRNA genes >3% difference. Pichler et al. (2018) USEARCH version 10.0.240 was used for quality

control and OTU picking,(Edgar, 2013) OTUs were clustered at 97% sequence identity. The taxonomic relationship of OTU representative sequences were identified by BLASTn searches against SILVA database (www.arb-silva.de) version 132. To

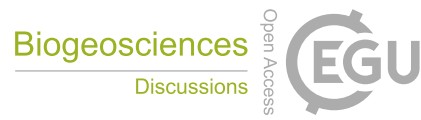

identify contaminants, 16S rRNA genes from extraction blanks and dust samples from the lab were also sequenced. These 16S rRNA gene sequences from contaminants were used to identify any contaminating bacteria in our samples. All OTUs containing sequences from these 'contaminant' samples (<5% of total) were removed prior to downstream analysis.

220    The qPCR and sequencing data were then used to quantify the abundance of individual 16S rRNA genes per OTU across the sampled water column, in the different biogeochemical zones. The fractional abundance (percent total sequences per sample) of each 16S OTU was multiplied by the total number of 16S rRNA genes per sample. This provided quantitative gene abundance per OTU, converting the relative abundance in the 16S rRNA gene libraries into quantitative values.

## 225  3  Results

### 3.1 Temperature, sulfate and DOC depth-profiles

DOC concentrations were highest at the lake surface with concentrations of nearly 5 mg/L and decreased to values of around 3 mg/L at the lake bottom (Fig. 1A). The lake water temperature was 18°C and decreased to 5°C at a water depth of around 12m (Fig.1A). As a result, summer warming resulted in a stratification of lake Fohnsee with the development of an anoxic

hypolimnion between 12 m and 22 m from around May to September with a constant temperature of 5°C.
Sulfate concentrations were 9.5 mg/L in the epilimnion and remained unchanged within the analytical uncertainty in the anoxic hypolimnion (Fig. 1A). Sulfate concentrations only decreased from a mean value of around 9.5 mg/l to 6.7 mg/l very close to the water/ lake- sediment interface.

### 235  3.2 Depth-profiles of $O_2$, $NO_x^-$, $NH_4^+$, and stable water and nitrogen isotopes

Aerobic redox conditions were prevalent within the epilimnion with a steep oxygen concentration gradient from > 9 mg/L at the surface towards < 0.02 mg/L below 12 m (Fig. 1B). The average concentrations of nitrate in the epilimnion was 6.7 mg/L (Fig. 1B). Below 12 m, in the nitrate-methane transition zone (NMTZ), dissolved oxygen concentrations decreased below detection (< 0.02 mg/L) and at a water depth of 21 m nitrate concentrations had decreased to < 0.15 mg/L, while nitrite

concentrations peaked at 1 mg/L at a water depth of 20 m. Ammonium concentrations decreased from around 1 mg/ L at the lake bottom to the oxycline and were below detection (< 0.15 mg/L) above a water depth of 12 m (Fig. 1B).

**Figure 1A and 1 B**

$\delta^{15}N$ and $\delta^{18}O$ values of dissolved nitrate increased in the anoxic water column ($O_2$ concentration < 0.02 mg/L at a water depth below 12 m) of the lake from 6.7‰ to 45.4‰ for $\delta^{15}N$ and from around 1.7‰ to 21.5‰ for $\delta^{18}O$ (Fig. 2C). Simultaneously, $\delta^{15}N$ of nitrite increased from 0.1‰ to 25.9‰ concurrently with increasing $\delta^{15}N$ values of nitrate, while $\delta^{18}O$

values of nitrite remained quite constant over the water depth (Fig. 2C). The $\delta^{15}$N values of ammonium increased from 7.9‰ at the lake bottom to 11.6‰ near the NMTZ, while simultaneously decreasing ammonium concentrations were observed

(Fig. 2).

The oxygen isotope ratios of lake water varied between -10.4 and -9.5‰ for $\delta^{18}$O. $\delta^2$H values were -73.0‰. In the aquifer the $\delta^{18}$O value was very close to -10‰ (Fig. 1B) supporting earlier findings that lake water is mainly derived  from groundwater (Braig et al., 2010).

**Figure 2A to C**

**3.3 Depth-profile of methane concentrations and C isotope ratios**

Concentrations of dissolved methane were highest in the methane zone (from 22 m to 20 m) near the lake bottom with concentrations of 2.5 mg/L but decreased to concentrations below the detection limit towards the NMTZ (from 20 m to 12 m).

With decreasing methane concentrations, $\delta^{13}C_{CH4}$ values increased from -72 ‰ at the lake bottom to -39 ‰ at a water depth of 18 m in the NMTZ (Fig.2B). Above a water depth of 18 m, methane concentrations were too low to analyze the stable carbon isotope ratios of methane. The steepest counter-gradients of nitrate and methane concentrations were observed at a water depth between 18 and 21 m (Figs. 2A and 2B), exactly, where nitrite concentrations peaked (Fig. 2A).

**3.4 Microbial community distribution in the water column of Lake Fohnsee**

To support our interpretation of chemical and isotopic data we determined the abundance and distribution of key microbial groups involved in N and C cycling processes and performed high-throughput Illumina sequencing of the V4 hypervariable region of the 16S rRNA genes together with quantitative PCR (qPCR). Analysis of similarity (ANOSIM) performed on the data revealed that significantly (R: 0.57, $P$ = 0.002) different microbial communities inhabited four geochemical zones in the

water column, the oxic lake water (6 m), the upper NMTZ (12-14 m), the lower NMTZ (16-18 m), and a methane rich zone near the lake bottom, where nitrate and nitrite concentrations decreased towards to the detection limit (20-22 m) (Figs 2A and 3). The differences in the communities are attributed to a decrease in the *Verrucomicrobia* and *Actinobacteria* with depth, and a large increase in the relative abundance of *Gammaproteobacteria* at a water depth of 22 m (Fig 3B). While present at a lower relative abundance, *Epsilonproteobacteria*, *Deltaproteobacteria*, and *Bacteroidetes* also increased with increasing depth

below the oxycline (Fig 3B).

The relative abundance of populations (operational taxonomic units sharing 97% sequence identity) was converted into quantitative terms by multiplying the fractional (relative) abundance of the populations against the total number of 16S rRNA





gene copies per sample.  This revealed a peak in microbial abundance just below the oxic-anoxic transition zone between 12 and 14 m, as well as the presence of known operational taxonomic units (OTUs) affiliated with anaerobic methane oxidizers

(*Crenothrix*, NC10) and potential OTU affiliated with '*Candidatus* Anammoximicrobium' (Fig. 3C).  The methane oxidizing *Crenothrix* and NC10 OTUs showed peak abundance below the oxic – anoxic transition zone at 12-14 m, whereas the *anammox* bacteria '*Candidatus* Anammoximicrobium' showed peak abundance in this zone and in the deeper water zone between. 20 and 21m (Fig. 3C).

**Figure 3**

## 4    Discussion

### 4.1  Evidence of AOM coupled with denitrification in the nitrate-methane transition zone (NMTZ)

To test the hypothesis if methane diffusion from the lake sediments towards the oxycline or micro-aerobic methane oxidation (Blees et al., 2014) can explain the methane concentration profile in the NMTZ, a simple 1D diffusion model with a constant methane input ($C_0$= 2.6 mg/L) was used (Fig. 4). Subsequently, the diffusion model was extended with a degradation term and the oxygen concentration profile from 12 m below lake surface to the lake bottom was modelled. A constant input oxygen concentration of 0.02 mg/L was used as a bounding condition, as this value is the detection limit of the oxygen sensor used in

this study. Furthermore a maximum reduction rate for oxygen of 0.07 $d^{-1}$ was used that represents a value that is at the lower boundary of literature data (Blees et al., 2014; Zimmermann et al., 2019). The results provide clear evidence that diffusion controlled methane fluxes from the sediment surface to the oxycline are highly insufficient for explaining the non-linear decrease of methane concentrations in the water column. To test the hypothesis whether micro-aerobic methane degradation is a driving process in the hypolimnion of Lake Fohnsee, a model run with a 1D diffusion model linked with a very small

degradation term (k= 0.07 $d^{-1}$) was performed. Results showed that oxygen concentration decreases from 0.02 mg/ L at a water depth of 12 m to < 0.002 mg/L at a water depth below 16 m and suggest that only a very small fraction of methane can be oxidized with trace amounts of oxygen. Hence a different process is required to explain the methane concentration profile in the NMTZ below 12 m (Fig. 4).

**Figure 4**

The vertical distribution of electron acceptors in the water column of lake Fohnsee was in agreement with the expected order of decreasing free-energy yields (Appelo and Postma, 2005). Nitrate concentrations decreased in the water column at a depth below 12 m, where model results suggest that dissolved $O_2$ was available in trace amounts (Fig. 1B). Sulfate concentrations of





around 9.5 mg/L remained unchanged throughout the water column in the presence of nitrate. Near the water – sediment

interface sulfate concentrations decreased slightly (Fig.1A).

Decreasing nitrate concentrations in the water column suggest that this electron acceptor may be microbially reduced in the

anoxic water column of the lake coupled with the oxidation of DOC or methane that are both present in Lake Fohnsee water

(Figs. 1 and 2B). Stable isotope technique was used to test the hypothesis whether denitrification occurred and was coupled

with methane as an electron source.

Methane is formed by methanogenesis in the sediments (Conrad et al., 2007; Norði et al., 2013) and diffuses upwards toward

the oxycline. The $\delta^{13}C$ values of -71.6‰ for dissolved methane at the bottom of lake Fohnsee (Fig. 3B) indicate a biogenic

source (Norði et al., 2013; Rudd and Hamilton, 1978). In absence of dissolved oxygen (< 0.02mg/L), methane concentrations

decreased and $\delta^{13}C$ values of methane increased to values of -38.6‰ toward the oxycline (Fig. 2B), providing evidence for

AOM. At this depth interval, nitrate concentrations also decreased and $\delta^{15}N$ and $\delta^{18}O$ values of nitrate increased from around

5‰ to 45‰ and from around 1‰ to 22‰ (Fig. 2C), respectively, while nitrite concentrations peaked (Fig. 1B). This provide

clear evidence that denitrification occurred. Therefore, the chemical and isotopic data clearly demonstrate that microbial nitrate

reduction and AOM has occurred concomitantly in the NMTZ between a water depth of 16 and 20 m at Lake Fohnsee.

**4.2 Evidence of anammox at the bottom of the NMTZ**

We also present several lines of qualitative and quantitative evidence that *anammox* has occurred with *AOM* coupled with

denitrification at the bottom of the NMTZ of the lake. As expected the nitrite concentration at a water depth of 20 m was

highest where nitrate reduction occurred (Fig. 2A). Between this depth and the lake bottom, our data strongly suggest that

*anammox* is the main sink of $NH_4^+$. Ammonium occurs in concentrations of up to 1 mg/L at the bottom of the water column

at 22m, likely stemming from the heterotrophic degradation of organic nitrogen (e.g., proteins and amino acids) close to the

sediment – water interface, and is subsequently transported from the methane zone near the lake sediments into the overlying

water column (Norði et al., 2013; Wenk et al., 2014), where the $NH_4^+$ concentration decreases continually towards < 0.15 mg/L

at 12 m depth. The decrease in ammonium concentration with increasing water depth is accompanied by an enrichment of $^{15}N$

in the remaining ammonium shifting the $\delta^{15}N_{NH4}$ values from 7.9‰ to 11.6‰ between 22 and 20 m water depth (Fig. 2C),

suggesting that ammonium is oxidized anaerobically while enriching the remaining substrate in $^{15}N$. Above this water depth

there is no isotopic evidence that ammonium is oxidized under anaerobic conditions and the decrease of ammonium

concentrations may be only affected by diffusion and by ammonification and nitrification processes that may occur at the

oxycline. A difference of $\delta^{15}N$ values ($\Delta$ $\delta^{15}N$) of nitrate and nitrite of around 11‰ was observed in the NMTZ at depths of

16 and 18 m, where *AOM* coupled with denitrification controls microbial nitrate reduction. When new nitrate is formed as

metabolic product by nitrite oxidation during *anammox*, the $\delta^{15}N$ value of the newly formed nitrate is affected by an inverse

isotope effect (preferential removal of $^{15}N$ from the nitrite pool during oxidation to nitrate) resulting in nitrate that is strongly





enriched in $^{15}$N by up to 61‰ relative to the $\delta^{15}$N value of nitrite (Brunner et al., 2013). In this study the difference between $\delta^{15}$N values of nitrate and nitrite ($\Delta\delta^{15}$N) increased from 11‰ in NMTZ to > 26‰ at the water depth of 20 m, where $\delta^{15}$N values of ammonium increased while $NH_4^+$ concentrations decreased (Fig. 2C). This strongly suggests that the additional

isotopic difference in $\delta^{15}$N values between nitrate and nitrite of around +15‰ is likely the result of production of highly $^{15}$N enriched nitrate derived from *anammox*. The reason for the observed isotopic differences between nitrite and nitrate ($\Delta\delta^{15}$N) during the *anammox* process within in this study ($\Delta\delta^{15}$N of +26‰) compared to the results ($\Delta\delta^{15}$N of +61‰) found in a laboratory experiment (Brunner et al. 2013) could be the result of different *anammox* strains in lake water and the microcosm-experiment, limiting environmental concentrations of nitrite or that the suggested inverse isotope effect by *anammox* was

superimposed by "normal" isotope effects on denitrification in the lake water at a water depth of 20 m.

Furthermore, the deviation of the slope of $\delta^{18}$O versus $\delta^{15}$N values on a dual isotope plot (2D plot) for nitrate from the expected value of 1 for microbial denitrification (Knöller et al., 2011; Wunderlich et al., 2012) can be used to identify *anammox*. Granger and Wankel (2016) used a modelling approach linked with pH-dependent isotope exchange reactions between water-oxygen and nitrite-oxygen (Buchwald and Casciotti, 2010; Casciotti et al., 2007; Casciotti et al., 2010) to demonstrate that in a $\delta^{18}$O

vs. $\delta^{15}$N plot for nitrate a slope lower than 1 is a powerful indicator for the occurrence of *anammox* in an anoxic environment. During *anammox* when nitrite is reduced with ammonium as electron donor to nitrate, one oxygen molecule from water with a $\delta^{18}$O value of around -10‰ is incorporated into the newly formed nitrate. Additionally, $\delta^{18}$O values of nitrite are lowered due to rapid oxygen isotope exchange with water oxygen (Buchwald and Casciotti, 2010; Casciotti et al., 2007; Casciotti et al., 2010) and remained, therefore, constant over the water depth of 16 m to 20m (Fig. 2c). As a result, the *anammox* process

facilitates that $\delta^{18}$O values of nitrate remain low, while $\delta^{15}$N of the remaining nitrate is affected by an inverse nitrogen isotope effect. The $\delta^{18}$O vs. $\delta^{15}$N plot for nitrate samples from depths between 20 and 22 m in our study displays a slope of 0.5, while the slope was 0.8 in the NMTZ between 20 and 12 m, much closer to the typical trajectory for denitrification of ~1 obtained under laboratory experiments (Fig. 5). The much slower slope of 0.5 on the $\delta^{18}$O vs. $\delta^{15}$N plot for nitrate is an additional line of evidence that strongly suggests that *anammox* occurred at the bottom of the NMTZ between 20 m and 21 m.

**Fig. 5**

### 4.3 Crenothrix, NC10 bacteria and annamox bacteria in the water column of lake Fohnsee

We identified gamma-proteobacterial methane oxidizing bacteria related to *Crenothrix* that reach their peak abundance particularly in the NMTZ of the water column of the lake (between 12-20m). The abundance of *Crenothrix* rRNA gene copies

reaches up to $10^5$ (Fig. 3B), which is 2-3 orders of magnitude higher biomass reported for *Crenotrhix* in the Swiss alpine Lake Rotsee and Lake Zug (Kits et al., 2015; Oswald et al., 2017), where they may act as denitrifying methanotrophs that also have the capability for aerobic metabolism. The facultative metabolism of *Crenothrix* likely allows them to adapt to changing environmental conditions and may outcompete the nitrate reducing ANME-2d with lower doubling times in the denitrification





zone of stratified lakes (Deutzmann et al., 2014). We did not detect any representatives of the ANME-2d in our 16S dataset –

despite relatively deep sequencing depth (>150 000 reads per sample), indicating that if they were in the lake water, they were at abundances below our detection limit. ANME-2d may, therefore, be major contributors to AOM in bioreactor studies (Haroon et al., 2013; Shen and Hu, 2012) and sediments but not in the water column of this lake.

The presence of two separate populations of NC10 bacteria at a water depth between 12 and 22 m, in the region where also anaerobic oxidation of methane linked with denitrification exists, may suggest that this organism was also partially contributing

to the anaerobic oxidation of methane with nitrite (*n-damo*). However, again it remains unclear whether *Crenothrix* that also peaked in this region completely reduced dissolved nitrate to $N_2$ or both, NC10 bacteria ($NO_2^-$ reduction) and *Crenothrix* are involved in the N loss processes in potion of the water column. Since the highest abundance of NC10 bacteria in our and other studies is often observed at the oxic - anoxic interface (Ettwig et al., 2008), it is controversially discussed whether *M. oxyfera* can also use external $O_2$ to oxidize methane near the oxycline. Therefore, the respective roles of NC10 and *Crenotrix* remains

unclear in this study.

Within the anoxic regions of the water column (NMTZ and methane zone), the OTU affiliated with '*Candidatus* Anammoximicrobium' is ubiquitous (Fig. 3B) and its lack of detection in the oxic zone indicates that it is a strict anaerobe. '*Candidatus* Anammoximicrobium' is an aggregate forming bacterium corresponding to a new genus within the Planctomycetes that is capable of anaerobically oxidizing ammonium with nitrite, and has been previously found to carry out

*anammox* in a wastewater bioreactor (Khramenkov et al., 2013). The '*Candidatus* Anammoximicrobium' and NC10 bacteria both utilize nitrite as a terminal electron acceptor, and they co-occur at depth of 20 m, where highest nitrite concentrations were observed (Fig. 2B). While activity indicators such as transcriptomes or NanoSIMS are needed to prove the *anammox* activity of *Candidatus* Anammoximicrobium' in our samples, the stable isotope and geochemical profiles indicate that this OTU is present in a geochemical setting where *anammox* is taking place. This, together with its affiliation to '*Candidatus*

Anammoximicrobium', indicates that this OTU has the potential to perform *anammox* in the aquatic environment of Lake Fohnsee at a depth of 20 m. In the water depth where nitrite was available due to denitrification via anaerobic methane oxidation*, both *anammox* and NC10 bacteria could compete for the same available nitrite as speculated for *Crenothrix* and NC10 bacteria in the NMTZ as sown in Fig. 6.

The detected microorganisms at Lake Fohnsee were found to be ecologically important in driving the C and N cycles of other

stratified lakes (Deutzmann et al., 2014; Naqvi et al., 2018; Oswald et al., 2017). This make it highly likely that these microbial groups are also potentially responsible for the removal of nitrogen and methane at Lake Fohnsee.

**Figure 6**


### 5    Conclusion



While aerobic methane oxidation has been proven to occur for over a century in lake water, knowledge on *anammox* and AOM coupled with denitrification in the same natural environment is scarce. Our field study results show, that *AOM,* denitrification and *anammox* are potentially coupled at Lake Fohnsee as previously demonstrated in bioreactor studies for *n-damo* and
*anammox* (Ettwig et al., 2010; Haroon et al., 2013). The suggested linkage of the N and C cycles at Lake Fohnsee is a previously overlooked process in stratified lakes that have the potential to contribute to the removal of nitrate, nitrite and ammonium from aquatic ecosystems by converting them to harmless $N_2$ while oxidizing the potent greenhouse gas methane to highly soluble $CO_2$.

**Data availability**

Illumina sequencing data for community analyses are deposited at NCBI BioSample (www.ncbi.nlm.nih.gov/biosample) under accession number PRJNA541816.

**Author contribution**

FE has designed the study, AW has performed the field work and the measurements of stable water isotopes. Instrumentation and methodology were provided by MS for N isotopes and BM for $CH_4$, FE has performed the modelling study, ÖC and WO have performed the qPCR, whereas WO has interpreted the data, FE, AW, BM, MS and WO have discussed the results, and FE wrote the original manuscript supported by BM and WO, whereas AW has visualizied the isotope and water chemistry data.
**Acknowledgement**

We thank S. Thiemann for technical assistance and the staff of the Research Station Iffeldorf for the support during field-work.

**Financial Support**

Funding was provided by DFG grant EI 401/10-1 to F. Einsiedl and by a NSERC discovery grant to B. Mayer. Illumina
sequencing data for community analyses are deposited at NCBI BioSample (www.ncbi.nlm.nih.gov/biosample) under accession number PRJNA541816.

**Conflict of Interest**

The authors declare no conflict of interest.

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





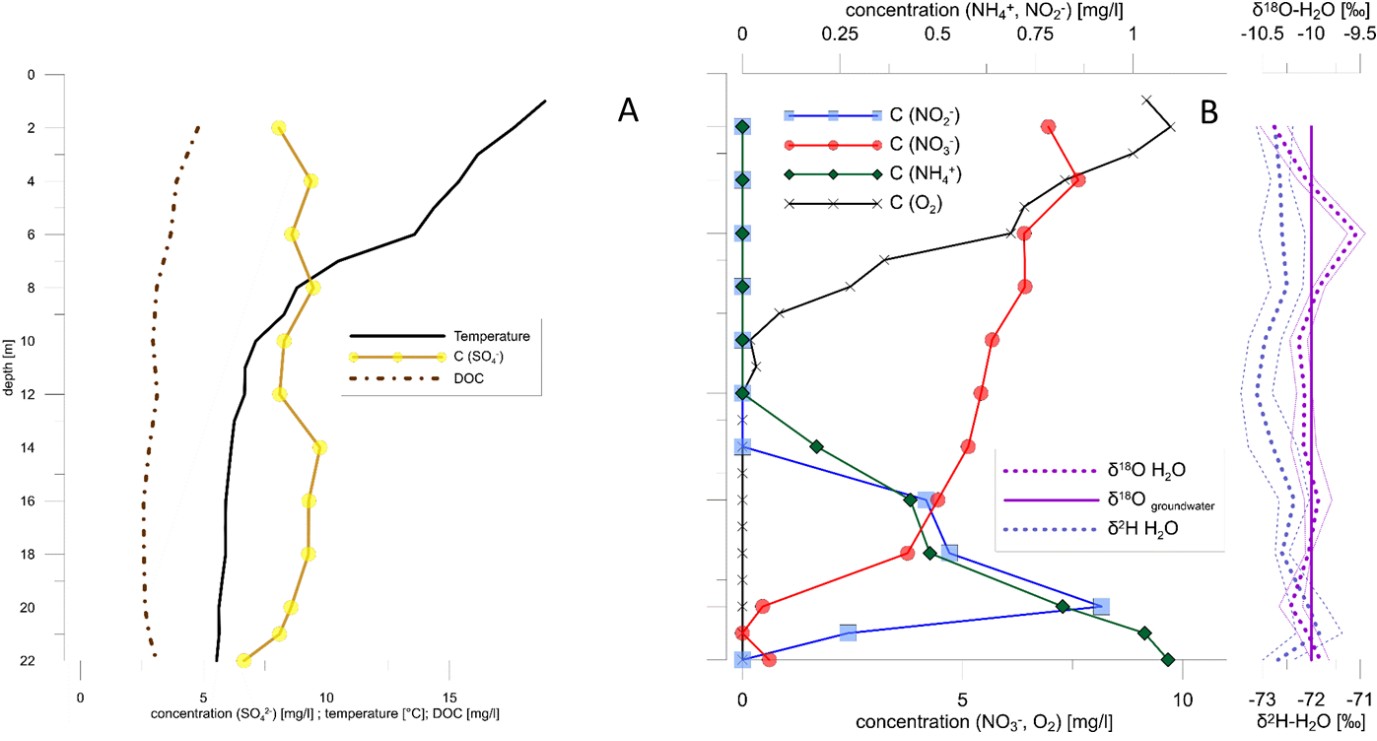

**Figure 1A and B. Temperature profile and vertical distribution of concentrations of DOC, sulfate (left) and (B) dissolved nitrate, nitrite, ammonium, and dissolved oxygen, $\delta^2H$ and $\delta^{18}O$ values of lake water and $\delta^{18}O$ value of groundwater (right).**


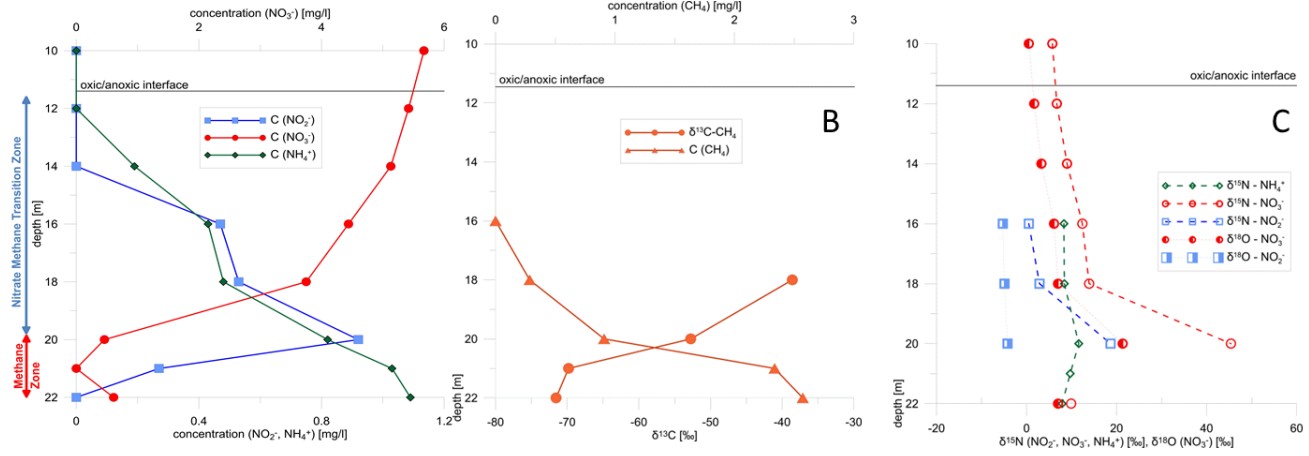


**Figure 2A-C. Water column profiles below the oxycline (10 to 22 m) for methane, nitrate, nitrite and ammonium concentrations and (B) stable isotope data and concentration profile of methane ($\delta^{13}C$), and (C) nitrate ($\delta^{15}N$, $\delta^{18}O$), nitrite ($\delta^{15}N$, $\delta^{18}O$), and ammonium ($\delta^{15}N_{NH4}$) isotopes.**







Figure 3A-C. Analysis of 16S rRNA gene data from microbial communities in the stratified lake. (A) Heatmap showing the relative abundance of specific groups in the 16S rRNA gene sequencing data, and corresponding heirarchical clustering analysis (analysis of similarity (ANOSIM) P value = 0.002) of four geochemically defined zones. For those depths where replicates were obtained, the data for both replicates are shown.

(B) The relative abundance of 16S rRNA gene sequences affiliated with the major groups across the stratified water column. (C) Abundance of 16S rRNA gene copies determined via qPCR, and the qPCR normalized absolute abundances of 16S rRNA gene sequence relative abundances from key populations (OTUs) potentially involved in AOM and anammox, specifically those affiliated with Crenothrix, NC10, and potential anammox bacteria.



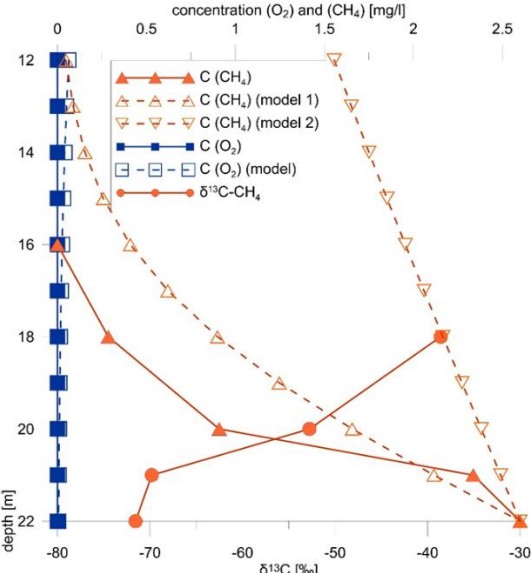

**Figure 4. Depth-profiles of methane concentration (filled triangles and its isotopic composition (filled circles) within the water column, modelled methane concentrations (open triangles) using a 1D-diffusion model and a diffusion coefficient for $D_{CH4}$ of 2.1 (model 1) and 0.1 m² day⁻¹ (model 2), measured and modelled oxygen concentrations (open and filled squares, respectively) where the 1D diffusion model was additionally linked with a degradation term (first order rate constant k= 0.07 d⁻¹, diffusion coefficients of $D_{O2}$= 0.2 m² day⁻¹).**





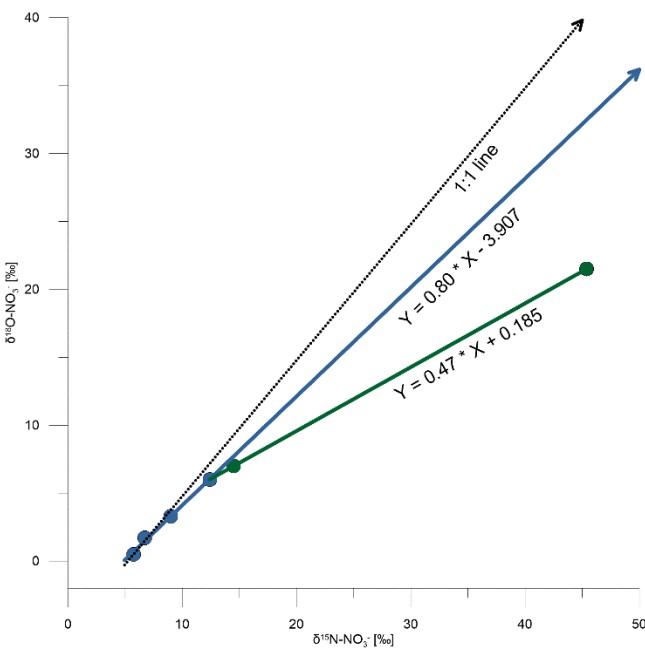


**Figure 5. δ¹⁸O versus δ¹⁵N plot of nitrate with the typical trajectory of 1 for denitrification obtained under laboratory conditions (black line), calculated trajectory of 0.8 for the *n-damo* zone (20m and above, blue line) and around 0.5 for the *anammox* zone (20-22m, green line).**

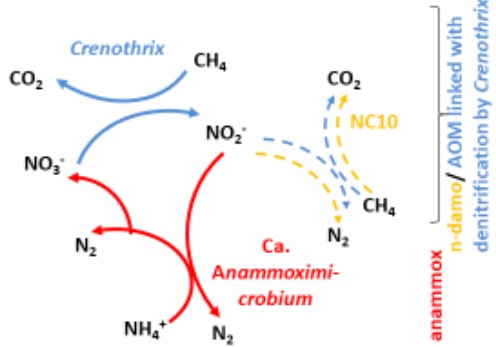


**Figure 6. Conceptual model of the coupled N and C cycles in the anoxic water column of Lake Fohnsee**