# Peer review of "Biogeochemical evidence of anaerobic methane oxidation and anaerobic ammonium oxidation in a stratified lake using stable isotopes"

_Biogeosciences, 2020_

## Referee Comment (RC1) · Anonymous Referee #1 · 25 Mar 2020

Review of "Biogeochemical evidence of anaerobic methane oxidation and anaerobic ammonium oxidation in a stratified lake using stable isotopes" by Einsiedl et al.

Summary

The authors present a comprehensive dataset highlighting key redox transformations of carbon and nitrogen in a stratified lake in southern Germany. The study takes advantage of patterns of stable N, O and C isotope fractionation together with changes in concentrations to deduce activity of microbial catalyzed transformation pathways. Among the conclusions drawn from these data – the authors suggest the strong likelihood for biogeochemical linkages between and among denitrification, anaerobic methane oxi-

dation and anaerobic ammonium oxidation (anammox). Such linkages have not been widely demonstrated or observed in field settings, making this study relevant for understanding the prevalence of such cryptic connections across a range of other environmental settings.

Major comments

Overall, I found the study to be well executed and the data to be of high quality. The inclusion of the microbial community analyses offers a great complementary perspective on the purported geochemical connections drawn from the isotopic and concentration analyses. The overlapping zonation of several key redox-active species in this vertical profile of a lacustrine system provides compelling evidence for their putative involvement with one another. However, compelling evidence is not always enough to draw definitive conclusions about the proposed interactions. In this respect, the authors seem well aware of the limitations of their data, and temper their conclusions with an appropriate amount of caution. In some cases, however, I still feel that there can be slight over-reach (e.g., L339), in concluding that direct coupling of denitrification and anaerobic methane oxidation was demonstrated, for example. In most cases, demonstrating the environmental coupling of reactants requires careful ex-situ experiments, with manipulations and controls. The authors are leading experts in the field of microbial and (isotope) biogeochemistry – so they understand these aspects very well. I would just recommend that they pay special attention throughout the manuscript to state (which they mostly do already) that their data provide compelling evidence for – or suggest the existence of these linkages (without needing to claim that their data prove it).

In general, there are some sentences that suffer from awkward wording. I made suggestions to help smooth these sections as outlined below. I recommend publication after the authors address my (mostly minor) comments as outlined below:

1. The quality of the figures seems sub-par and some effort should be given to improve

on details such as text and symbol sizes and colors, axes labels and ticks, etc.

2. In general, I find the choice of units (e.g., mg/L) problematic (not knowing whether mg of N or mg of NO3- are being presented, for example). I would strongly suggest expressing concentrations in the more widely understood values of molarity (e.g., micromolar). This also has the advantage of facilitating more direct comparisons (at molar stoichiometries) among N, C, O and S cycling, etc. – thereby helping readers better understand the arguments being presented.

3. L 137: What is the reasoning behind the two diffusion coefficients for methane? This is presented in an apparent attempt to bracket a range of acceptable flux estiamtes, but is not explained in the text.

4. L131: Something about Equation 1 seems incorrect. Are there meant to be two equalities here? Do the 'x' terms both refer to depth? In general I think 'z' is more frequently used for referencing to depths. Please confirm that the expression of this diffusion equation is properly written.

5. L310: I understand that in the presence of NO3-, sulfate reduction is not thermodynamically predicted to proceed, however some arguments have been made for processes occurring in micro-zones inside of particles, for example. How much anticipated change in sulfate would be predicted – and would the IC measurements actually be sensitive enough to this? The increasing presence/abundance of Deltaproteobacteria also lend some credence to the idea that at least some level of $SO_4^{2-}$ reduction could be occurring. Changing units into molarity would help readers with this comparison as I noted above.

6. The $\delta 18O$ values of nitrite are reported, but nowhere in the text is it explained how these values were determined or calibrated. Further, given the low pH of lake water, the $\delta 18O$ values of nitrite are very likely to be in isotopic equilibrium with the water, yet appear to fall around -4 to -6‰ which would be much too low. Given a lake water $\delta 18O$ value of $\sim$ -10‰ – the $\delta 18O$ value for nitrite in equilibrium should fall closer to

+4‰ (see Casciotti et al., 2007). Finally, the $\delta18O$ values of nitrite in this study are not mentioned or involved in aspect of the conclusions – and should probably be omitted for clarity (e.g., they aren't used to bring any new insight into the system as presented).

Minor comments

L15: "Nitrate dependent anaerobic methane oxidation and anaerobic oxidation of ammonium (anammox) have the potential..."

L20: anammox does not require italics.

L24: Perhaps consider using " is the most parsimonious explanation"

L32: ... I recommend: "... contain significantly different microbial communities that include bacteria known to be involved in..."

L47: ...coupled to nitrate...

L54: ... ANME-2d lineage promotes/conducts the reduction of nitrate...

L57: Besides

L59: ...related to Crenothrix also have the...

L60: Crenothrix is misspelled.

L61: ... may act as a driver...

L74: references do not need to be italicized

L86: ...sequencing of 16S rRNA genes that provides evidence for the...

L115: ... sterile filter, which was then kept frozen...

L149: ... which represents the lower bound...

L153 – 157 – this should be moved to introduction or discussion.

L168: ...(2007), respectively.

L168: Nitrite was converted to N2O using acetic acid buffer sodium azide.

L169: . . . mixture of both nitrate and nitrite was reduced to N2O via azide.

L171: Can you provide some estimate of error propagation for this inverse mixing calculation?

L173: . . . by buffered azide solution for subsequent analysis.

L181: How were the $\delta$15N values of the nitrite standards determined – and to what level of precision? There is no mention of calibrated $\delta$18O standards for nitrite. Yet $\delta$18O of nitrite data are reported (albeit not discussed). Please clarify or omit.

L192: Here it is unclear if the methane isotope analyses were conducted on the same bottles? Via manual injection? Was this a full bottle purge and trap approach? Was this automated? Were there standards included in this approach? How were the analyses standardized (e.g., extractions of methane of known composition from water?)?

L236: Aerobic conditions. . .

L237: The average concentration of nitrate. . .

L239: . . .nitrate concentrations decreased. . .

L261: . . . were too low for stable isotope analyses.

L298-300 – Rephrase. Awkward wording.

L301:. . . water depth below 16m, suggesting that only a very small fraction of methane can be oxidized with such trace amounts of oxygen.

L314: A stable isotope technique was used. . .

L327: Awkward sentence. . . perhaps reconsider phrasing "We also present several lines of qualitative and quantitative evidence for the co-occurrence of anammox together with denitrification coupled to AOM towards the bottom of the NMTZ."

L339: Here the language reads as though AOM coupled to denitrification has been unequivocally demonstrated, which isn't exactly the case. I think here it is best to qualify this a bit more. In fact denitrification could also easily be coupled to more canonical heterotrophy at the same time.

L344: Again – I would recommend softening and rephrasing: "This is consistent with the additional isotopic difference in $\delta$15N values between nitrate and nitrite of around +15‰ arising as the result of production of highly 15N enriched nitrate deriving from anammox."

L350: . . .superimposed on 'normal' isotope effects. . .

L356: As written this statement is incorrect. I would recommend restating: "During anammox, when nitrite is reduced with ammonium as electron donor and nitrate is produced, one oxygen atom from water having a $\delta$18O value of around -10‰ is incorporated into the newly formed nitrate." This incorporation of a new O atom is also most likely associated with a kinetic isotope effect – as has been demonstrated in nitrite oxidizing bacteria (see Buchwald and Casciotti, 2010).

L359: . . . the anammox process leads to $\delta$18O values of nitrate remaining low, while. . .

L361: . . .by an inverse isotope effect and values continue to increase.

L368: Although not mentioned, I am curious whether any nitrite oxidizing bacteria were detected in the genomic analyses? I assume from their omission that they were not. This could be a useful fact to mention if so.

L373: . . . environmental conditions, helping any nitrate reducing ANME-2d (with lower doubling times) in the denitrification zone. . .

L380: . . . the meaning behind this sentence is unclear. . .

L382: . . . potion? portion?

L398: . . . as shown. . .

L400: . . . This makes it highly likely. . .

Figure 1: There is a mismatch between the ticks and y-axis labels on both panels A and B. Perhaps an indication of where the bottom of the lake is on this figure would be useful for context. In panel B, the choice of symbols for the $\delta$D and $\delta$18O of water are essentially indistinguishable. In general, the axis labels are too small and much of the text/labels are difficult to read.

Figure 2: All of the text, labels, and axes are too small to be read.

Figure 4: Caption is missing a parenthesis ('filled triangles').

———————————————————

---

## Referee Comment (RC2) · Anonymous Referee #2 · 23 Apr 2020

The authors present a comprehensive data set (at least in terms of the number of measured parameters) on concentration profiles and corresponding stable isotope compositions of CH4, nitrate, nitrite, and ammonium in the water column of a stratified lake, suggesting links between anaerobic oxidation of methane (AOM), denitrification, and anammox. They attribute a nitrate methane transition zone to nitrate-dependent methane oxidation, and claim, based on model fitting approaches, that the observed methane profiles cannot be explained by microaerobic methane oxidation, and that coupling to denitrification is most likely. I do not think that the combined isotope profiles are that compelling with regards to their interpretation, and the respective enrichments in 15N in the ammonium and nitrate pools at around 20m depths could also be due

to denitrification and ammonium oxidation in close vicinity. Similarly, several scenarios may be possible to explain the observed offset in D15N between nitrite and nitrate. Nevertheless, it is a nice data set, and the authors do a good job in trying to interpret the data set in an attractive way. Additional evidence comes from molecular data, confirming the presence of the microbial players that potentially perform anammox and nitrate-dependent AOM. But there are no additional conclusive evidence on coupled N reduction and CH4 oxidation, and a more quantitative assessment with regards to the importance of these microbes, as well as turnover rates is missing. Clearly, incubation experiments with different electron acceptors would help. As a consequence, it remains uncertain, which processes truly govern the observed isotope profiles and in turn it is difficult to understand how important these combined processes are in regulating methane emissions and nitrogen concentrations in the studied lake, let alone in other lakes. The authors mention the limitations of their approach, but still, I would advise a more cautious interpretation of the data. To me this looks like a quite common seasonally stratified lake with very typical seasonal biogeochemical dynamics and processes. In this regard, I am afraid that the claimed "unique biogeochemistry" is a bit of an overselling. Most importantly, as I will outline below, I felt that the modelling is not sound, and the interpretation of isotope profiles "by eye" in a system that may not be in steady state problematic.

Main points:

Abstract: In light of much more important N-loss reactions (denitrification anammox) I think it is an exaggeration to state that nitrate-dependent methane oxidation has the potential to reduce nitrate loading. It just does not happen, we know that. I doubt that AOM-denitrification-anammox process really is an overlooked process. . .it simply is less important than canonical denitrification and aerobic methane oxidation. There is no information on the site/lake. The name and location of the lake should at least be mentioned.

The introduction is good, and provides more or less up-to-date info on AOM and anammox in freshwater ecosystems. Maybe it needs to be distinguished better between studies that were conducted in the field versus those that are purely experimental. This is not completely clear for someone who did not read the papers.

L56: Why is nitrate reduction more important in lakes than nitrite reduction, just because there is more nitrate than nitrite? Nitrite is an intermediate and assuming that the most important N loss pathway is complete denitrification, nitrite reduction has to balance nitrate reduction, if nitrite does not accumulate.

L88: "...coupled the diffusion model with a degradation term to clarify the effect of dissolved oxygen on methane oxidation. The observed coupled process has the potential to constitute an important sink of dissolved nitrogen ($NO_3^-$, $NO_2^-$, $NH_4^+$) and methane ($CH_4$) in freshwater environments." What exactly is coupled? What coupled process are the authors referring to? This is not clear at this point of the article what they did in the model and how $O_2$ thresholds are integrated. Even if an explanation will follow in the method section, this needs to be clarified (or moved to the more detailed sections on the model parametrization).

Model: There is not enough explanation of the model. Obviously, it is not a real reaction-diffusion model, but it also is not just a diffusion model, right? What are the reaction parameters, how are they set? I am not an expert in modelling, but it remains unclear how the modelling works, apparently, a purely diffusive part and a reaction part is combined, but the coupling of the model components is unclear. Most importantly, how well constrained is turbulent diffusion? The results (modelled concentration profiles and isotope ratios in water column) will be highly sensitive to the choice of the D, and adopting values for D (by the way D is used usually for molecular diffusion only) from other lakes may not be appropriate. In fact, the authors seem to have a very limited knowledge of modelling turbulent diffusion in lakes. Firstly, it seems that their choice of what they call D (or K in the literature) is at least two orders of magnitude higher than would be expected for a stratified lake. They cite Oswald et al. from which D was adopted. But looking into the paper by Oswald, I saw that their choice of $K_z$

was 4x10-3 cm2 s-1, which corresponds to approx. 0.035 m2 d-1. If the authors really used D/Kz values between 0.1 and 2.1, their modelled concentrations will be way off. Finally, assuming different turbulent diffusion coefficients for O2 and CH4 is nonsense. Turbulent diffusion is not solute-specific (in contrast to molecular diffusion), it is a hydrodynamic property of the flow field. As for the first-order methane oxidation rate coefficient, how can the authors just assume a value adopted from other studies? This parameter will change significantly between ecosystems, and has to be estimated based on fitting of the model to the observational data.

Nitrate/nitrite isotope measurements: The authors write: Nitrogen and oxygen isotope ratios of nitrate were calculated by measuring nitrite alone as well as the mixture of nitrite and nitrate in a sample and using an inverse mixing calculation to determine the isotopic ratios of nitrate alone. First of all, there seems to be a duplication in this sentence. I think I understand what the authors did. They measured the isotopic composition of nitrite, and then the isotopic composition of the mixture. Based on mass balance calculation, they then calculate the isotopic ratios of nitrate alone. This works for d15N, but does it work for d18O? I am pretty sure that it does not. In a sample that contains nitrite and nitrate, O isotope fractionation during the conversion to N2O is different for nitrite and nitrate. Hence the d18O of the N2O cannot simply be standardized, because the O-isotope offsets will be different for nitrite and nitrate. In other words, the d18O of a NOx sample is probably meaningless, and so will be the calculated d18O nitrate values. The nitrate d18O should have been measured after removal of the nitrite. Could the changes in Dd15N(nitrate-nitrite) be an artifact that is simply the result of this effect and changing nitrite/nitrate concentration ratios?

L188: How was complete outgassing of CH4 assured before headspace analysis? Was brine/NaOH added? Concentrations were calculated based on Henry's Law, but what about the d13C? Is there an isotope shift between CH4 in the headspace and the CH4 dissolved? If so, was that considered?

Results: I am a bit disappointed by the low number of data points/analyses. As a

consequence, isotope gradients are not well resolved (and their interpretation is hence complicated), and the profiles are not replicated for several time points. Do the authors assume steady state conditions? How relevant is this for the model fitting?

Figure 3c is very difficult to read? Why not showing profiles (connected symbols) for the most relevant OTUs. It is almost impossible to see the vertical structure.

Discussion: It is not clear to me what the arguments are that allow the authors to exclude oxic methane oxidation. I agree that the concentration profile suggests reaction below the redoxcline, but you do not need to model this to come to this conclusion. At the same time, do the authors assume steady state? Apparently, the lake undergoes seasonal fluctuations, so that the curvature of the concentration profiles may represent a non-steady state, and its interpretation with regards to where reaction takes place and where not is biased.

Again, maybe it is my fault, but I got the feeling that, overall, the modelling is not sound (or that it is simply not the right model, comprehensive enough to realistically simulate the different interlinked processes), and so are the conclusions drawn from the modelling. But maybe I just did not understand it. For example, in L298-303: How was the model used to test whether microaerobic respiration plays a role or not? I am sure that, depending on the half-saturation constants, the diffusivity, and the rate coefficient, a model with solely microaerobic methane oxidation could generate the methane profile observed, while keeping the $O_2$ levels close to detection. But again, can steady state be assumed at all?

L319: From my understanding, the methane ïĄď13C increase below the redoxcline does not necessarily indicate AOM. I am not saying that it is not AOM that causes the increase in ïĄď 13C. But even if methane oxidation occurred only in a relatively thin layer further up in the water column, we would see a ïĄď 13C gradient. The authors should model not only the concentration, but also the different $CH_4$ species (i.e,ïĂãïĄď 13CH4), then they would probably see it.

Can the authors explain why a 90% decrease in ammonium is associated with a ïĄd'15N shift of only 4‰ L335-7: The authors say that above 20 m water depth, there is no evidence for ammonium oxidation. Why? Because the d15NH4 values do not increase? But they also do not increase much below that depth, where the authors suggest that anammox occurs. And most strikingly, the ammonium profile is essentially linear all the way up to the oxycline. To me this suggests that not much ammonium oxidation is taking place at this depth, and essentiall all NH4 is oxidized at the oxycline.

340-345: The authors cite the anammox isotope effect study by Brunner and colleagues. But they mix up equilibrium and kinetic N isotope effects between nitrite and nitrate. The inverse kinetic N isotope effect, which applies to active nitrate production from nitrite by anammox, is much lower than the -61‰ mentioned.

The authors should explain better why anammox could produce a d18O vs d15N NO3 relationship of 0.5. Is this slope consistent with nitrate production from nitrite with the incorporation of O atoms from water? Such slopes in d18O vs d15N NO3 plots have been observed in several ground/freshwater studies. Does this imply that in all these environments anammox was the main N-loss pathway?

What is the relative abundance of "normal" nitrate and nitrite reducers compared to NC10 and Crenothrix?

General: The paper is prepared with a certain degree of carelessness, with a lot of typos and sometimes odd wording.

Minor points: L32: geochemical L47: coupled L54: affects the reduction? The whole sentence, at least the final part, does not read well L84: Not overlapping isotope values but overlapping C isotope effects! L120: mgN? Otherwise it is surprising that the detection limit is the same for the different N-components. L262: counter-gradients? 314: What is the stable isotope technique? Simple isotope measurements? L372: Metabolism L382: portion

---

## Author Comment (AC1) · 15 May 2020

We like to express our deep gratitude for the detailed and constructive feedback from two reviewers. Below, we have provided a detailed point-by-point list of answers and replies to the comments and suggestions raised by the reviewers. We have made every attempt to address the excellent suggestions and the numerous highly valuable recommendations where appropriate, and have provided detailed responses and explanations below.

Reviewer #1 agrees with our assertion, that links between nitrate-AOM and anammox have not been widely demonstrated in the literature, and that our study is an important step in developing an environmental understanding of this process. Reviewer #1 found the study well executed and the data of high quality. Reviewer #2 was more critical, while also stating that the authors did a good job in interpreting their data in an attractive way.

Response to general statement:

Reviewer #1 mentioned that the authors seem well aware of the limitations of their isotope results and temper their conclusions with an appropriate amount of the limitations of the presented data (with very few exceptions where a slight over-reach of data interpretation can be identified). In contrast, Reviewer #2 asked for a more cautious interpretation of the data.

We have settled in the revised manuscript on a compromise approach that is primarily based on opinion of Reviewer #1 that our original conclusions were well tempered, while we have also made several text additions in the revised manuscript that caution against an over-interpretation of our findings (for example L. 339).

Point-by-point response:

**Point-by-point response:**

*Comments of Reviewer #1*

*The quality of the figures seems sub-par and some effort should be given to improve*

*on details such as text and symbol sizes and colors, axes labels and ticks, etc.*

**Response:** 1 & 2: We agree that an improvement of the quality of the figures is necessary and we also have changed the expression of concentrations to mmol/L.

*L137: What is the reasoning behind the two diffusion coefficients for methane? This is presented in an apparent attempt to bracket a range of acceptable flux estimates, but is not explained in the text.*

**Response:** The modelling was already discussed in detail with Reviewer #2 and we made every attempt to address both recommendations. In the revised manuscript we have calculated the Kz for Lake Fohnsee, so it is not necessary to show two diffusion coefficients.

*Something about Equation 1 seems incorrect. Are there meant to be two equalities here? Do the 'x' terms both refer to depth? In general I think 'z' is more frequently used for referencing to depths. Please confirm that the expression of this diffusion equation is properly written*

**Response:** In the revised manuscript, we have corrected the equation after Clark (1975), have extended the used equations for clarification (see Rev. #2), and define in the manuscript that "x" represents the depth.

*L310: I understand that in the presence of NO3-, sulfate reduction is not thermodynamically predicted to proceed, however some arguments have been made for processes occurring in micro-zones inside of particles, for example. How much anticipated change in sulfate would be predicted – and would the IC measurements actually be sensitive enough to this? The increasing presence/abundance of Deltaproteobacteria also lend some credence to the idea that at least some level of SO42- reduction could be occurring. Changing units into molarity would help readers with this comparison as I noted above.*

**Response:** Sulfate concentrations were clearly above the detection limit of the IC and we observed a decrease of sulfate concentrations from 8 mg/L at a water depth of 21 m to around 7 mg/L close to the lake. This 14% decrease in sulfate concentration with increasing depth could be interpreted to indicate partial bacterial sulfate reduction in micro-environments of particles near the lake sediment surface as suggested by the reviewer or alternately, by mixing effects between sulfate-free water from the sediments, where methanogenesis may occur, and lake water. As we also found nitrate concentration at the same depth (22 m) where we observed decreasing sulfate concentrations we can only speculate what processes control decreasing sulfate concentrations.

To address this, we revised the manuscript to the following new text at the end of § 4.1:

Decreasing sulfate concentrations at the bottom of the lake and nitrate concentrations at the same water depth of less than 1 mg/L can be explained by partial bacterial sulfate reduction in micro-environments of particles near the lake sediment surface (Bianchi et al. 2028, Nature Geoscience volume 11, pages263–268(2018)) or by mixing effects between sulfate-free water from the sediments, where methanogenesis may occur, and sulfate-containing lake water.

*The δ18O values of nitrite are reported, but nowhere in the text is it explained how these values were determined or calibrated. Further, given the low pH of lake water, the δ18O values of nitrite are very likely to be in isotopic equilibrium with the water, yet appear to fall around -4 to -6‰ which would be much too low. Given a lake water δ18O value of ∼ -10‰ – the δ18O value for nitrite in equilibrium should fall closer to +4‰ (see Casciotti et al., 2007). Finally, the δ18O values of nitrite in this study are not mentioned or involved in aspect of the conclusions – and should probably be omitted for clarity (e.g., they aren't used to bring any new insight into the system as presented).*

**Response**:

We used international nitrate standard with known isotopic composition ($\delta^{15}N$ & $\delta^{18}O$ values) and a lab-internal standard for $\delta^{15}N$ of nitrite but not for $\delta^{18}O$ of nitrite, while using the measurement gas $N_2O$. In the revised version of the manuscript, we will add the following information: "The isotopic composition of nitrite was determined using the azide method, similar to the analysis of nitrate. In order to ensure the proper reduction of nitrite to $N_2O$, in

addition to the samples, internal laboratory standards for KNO2 were analyzed in each batch (Lb1, $\delta 15N$ = -63‰ and Lb2, $\delta 15N$ = +2.7‰). Corrections of the raw $\delta^{15}N$ values were made based on the known values of the nitrate and nitrite standards.

With respect to the observed $\delta^{18}O$ values of nitrite, the paper by Casciotti et al. demonstrates perfectly that there is an isotopic exchange between oxygen of the water and oxygen of nitrite. Once this exchange is achieved, an isotopic equilibrium is established depending on the isotopic fractionation. This fractionation leads to significantly higher $\delta 18O$ values of nitrite compared to those of water. In Casciotti's study, the isotopic fractionation determined for freshwater is around -14‰. So based on the $\delta 18O$ of the water in this study close to -10‰, the expected $\delta 18O$ for nitrite should be +4‰, assuming there is only abiotic exchange. More recently, Sebilo et al. (2019) published a study based on isotope tracing during nitrite or nitrate reduction. This study revealed that the oxygen isotope shift was immediate and the authors attribute it primarily to denitrifying bacteria, given the rapidity of exchange. In this study, the $\delta 18O$ of the water was close to -10‰ and the $\delta 18O$ of the nitrite during its reduction was relatively constant, oscillating between 0 and -2‰, and hence displaying lower values than those expected with abiotic exchange alone.

The results obtained in the here discussed manuscript, with relatively constant $\delta 18O$ values for nitrite close to -5‰ indicate that an isotopic exchange occurred between the oxygen of the water and the oxygen of nitrite occurred, and that the latter was predominately controlled by biotic reactions. Moreover, since denitrification alone should have resulted in a $\delta^{18}O$ value of the nitrite between -2 and 0‰, this discrepancy seems to confirm that another biotic process is taking place.

**Minor revisions:**

L15: "Nitrate dependent anaerobic methane oxidation and anaerobic oxidation of ammonium (anammox) have the potential..."

**Response:** Here it is not clear what changes the reviewer would like to see. We suggest to change the sentence as follows:

Nitrate-dependent anaerobic oxidation of methane and anaerobic oxidation of ammonium (anammox) are two recently discovered processes in the nitrogen cycle that can reduce nitrogen loading of aquatic ecosystems and to reduce methane emissions to the atmosphere.

All other minor revisions focusing more or less on awkward wording were accepted and have greatly improved the manuscript that now reads as follows:

L20: anammox does not require italics.

**Response**: was changed

L24: is the most parsimonious explanation"

L32: … contain significantly different microbial communities that include bacteria known to be involved in…"

L47: …coupled to nitrate…

L54: … ANME-2d lineage promotes/conducts the reduction of nitrate…

L57: Besides

L59: …related to Crenothrix also have the…

L60: Crenothrix "was corrected".

L61: … may act as a driver…

L74: references do not need to be italicized "was corrected"

L86: …sequencing of 16S rRNA genes that provides evidence for the…

L115: … sterile filter, which was then kept frozen…

**Response**: …kept frozen with the filtered microbial biomass

*L149: … which represents the lower bound…*

**Response:** Here we have re-written the modelling part and detailed answers can be found in our response to Reviewer #2

*L153 – 157 – this should be moved to introduction or discussion.*

**Response**: As a short introduction to the stable isotope section, we are of the opinion that this text fit well in the current section, and hence have made no changes.

**Response:** The suggested improvements of the reviewer were accepted and we made the following changes in the revised manuscript:

L168: ...(2007), respectively.

L168: Nitrite was converted to N2O using acetic acid buffer sodium azide.

L169: ... mixture of both nitrate and nitrite was reduced to N2O via azide.

L171: Can you provide some estimate of error propagation for this inverse mixing calculation?

Please also compare answer to Rev #2 (on page 6)

The calculation of the isotopic composition is based on the measurement of the isotopic composition of $N_2O$ with an IRMS and the correction between the values obtained for the standards and the values measured by linear regression. For samples obtained from 14, 16, 18 and 20m depth, both nitrite and nitrate are present. However, taking into account the concentration ratios, the amount of nitrite represents at most 10% of the total concentration for the samples except for the 20m sample where the nitrite concentration is around 1 mg/L and the nitrate concentration is around 0.5 mg/L. For this point, taking into account the two molecules and calculating the nitrate $\delta^{18}O$ gives a value of 5.6‰ whereas it was 5.4‰ without correction.

L173: ... by buffered azide solution for subsequent analysis.

L181: How were the δ15N values of the nitrite standards determined – and to what level of precision? There is no mention of calibrated δ18O standards for nitrite. Yet δ18O of nitrite data are reported (albeit not discussed). Please clarify or omit.

**Response:** We have the revised the text as follows:

"The isotopic composition of nitrite was determined using the azide method, similar to the analysis of nitrate. In order to ensure the proper reduction of nitrite to $N_2O$, in addition to the samples, internal laboratory standards for KNO2 were analyzed in each batch (Lb1, δ15N = -63‰ and Lb2, δ15N = +2.7‰). Corrections of the raw $δ^{15}N$ values were made based on the known values of the nitrate and nitrite standards.

L192: *Here it is unclear if the methane isotope analyses were conducted on the same bottles? Viamanualinjection? Wasthisafullbottlepurgeandtrapapproach? Wasthis automated? Were there standards included in this approach? How were the analyses standardized (e.g., extractions of methane of known composition from water?)?*

**Response**: As stated in the original text, "the concentrations and carbon isotope ratios of dissolved methane in the lake water samples were determined using the static headspace equilibrium technique (EPA, 2002) where 10% of the water sample in the capped bottles was replaced with helium followed by outgassing of the dissolved gases in the water sample into the headspace for 1 h at 25ºC.

In the revised text, we have now clarified that:

- methane concentration and C isotope ratios were determined from the same bottle;

- that only 10% of the bottle content was replaced with an inert headspace gas;

- that this process was not automated;

- standardization of the measurements was accomplished as follows: Instrument stability and linearity was ensured by daily measurements of an in-house methane mix of 5% CH4 (balance helium). Carbon isotope analyses of methane were standardized by measurements of Isometric Instruments (Victoria, BC, Canada) gases containing methane with known □13C values including the following: B-iso1 (δ13C = -54.5‰, δ2H = -266‰), L-iso1 (δ13C = -66.5‰, δ2H = -171‰), and H-iso1 (δ13C = -23.9‰, δ2H = -156‰);

- standard solutions with dissolved methane of know isotopic compositions were not available;

**Response:** The improvements suggested by the reviewer were accepted and we made the following changes in the revised manuscript:

L236: *Aerobic conditions...*

L237: *The average concentration of nitrate...*

*L239: ...nitrate concentrations decreased...*

*L261: ... were too low for stable isotope analyses.*

*L298-300 – Rephrase. Awkward wording.*

L301:... water depth below 16m, suggesting that only a very small fraction of methane can be oxidized with such trace amounts of oxygen.

**Response:** Here we have decided to remove the $O_2$ calculations and will present the results by using a numerical model in a different manuscript.

**Response:** The suggested improvements by the reviewer were accepted and we made the following changes in the revised manuscript:

L314: A stable isotope technique was used...

L327: We also present several lines of qualitative and quantitative evidence for the co-occurrence of anammox together with denitrification coupled to AOM towards the bottom of the NMTZ.

L339: Here the language reads as though AOM coupled to denitrification has been unequivocally demonstrated, which isn't exactly the case. I think here it is best to qualify this a bit more.

**Repsonse**: In fact, denitrification could also be coupled to more canonical heterotrophy at the same time. Hence we revised the text as follows

New: …. where *AOM* may affect microbial nitrate reduction, although more canonical heterotrophy could also occur.

L344: *Again – I would recommend softening and rephrasing:*

Old: This strongly suggests that the additional isotopic difference in $\delta^{15}N$ values between nitrate and nitrite of around +15‰ is likely the result of production of highly $^{15}N$ enriched nitrate derived from *anammox*.

**Response:** This is consistent with the additional isotopic difference in $\delta^{15}N$ values between nitrate and nitrite of around +15‰ arising as the result of production of highly $^{15}N$ enriched nitrate deriving from anammox."

L350: ...superimposed on 'normal' isotope effects...

L356: *As written this statement is incorrect. I would recommend restating:*

 *"During anammox, when nitrite is reduced with ammonium as electron donor and nitrate is produced, one oxygen atom from water having a δ18O value of around -10‰ is incorporated into the newly formed nitrate." This incorporation of a new O atom is also most likely associated with a kinetic isotope effect – as has been demonstrated in nitrite oxidizing bacteria (see Buchwald and Casciotti, 2010).*

**Response:** We agree that our simplified statement was not correct since kinetic oxygen isotope fractionation was not considered. We have now revised this sentence as follows:

During anammox, when nitrite is reduced with ammonium as electron donor and nitrate is produced, one oxygen atom from water ($\delta 18O$ value of around -10‰) is incorporated into the newly formed nitrate, with an additional kinetic oxygen isotope effect (Buchwald and Casciotti, 2010).

**Response:** The following suggested improvements of the reviewer were accepted and we made the following changes in the revised manuscript:

L359: ... the anammox process leads to $\delta 18O$ values of nitrate remaining low, while...

L361: ...by an inverse isotope effect and values continue to increase.

*L368: Although not mentioned, I am curious whether any nitrite oxidizing bacteria were detected in the genomic analyses? I assume from their omission that they were not. This could be a useful fact to mention if so.*

**Response:** Because nitrate and nitrite reduction is such a widespread trait held by many facultative anaerobic bacteria, it is not possible to use our 16S rRNA gene sequence data to specifically show the abundance of 'normal nitrate and nitrite reducers' as the reviewer suggested. However, the Gammaproteobacteria are very abundant in our samples, and are well known to have many species that are capable of nitrate and nitrite reduction, a trait that is widespread throughout this class. Since the Gammaproteobacteria relative abundance increases with depth into the anoxic zone (Fig. 3b), it is likely that many of the Gammaproteobacteria in deeper waters of the lake are responsible for nitrate and nitrite reduction, and denitrification.

We will add a few lines to the revised manuscript.

L373: ... environmental conditions, helping any nitrate reducing ANME-2d (with lower doubling times) in the denitrification zone...

Was accepted

L380: ... the meaning behind this sentence is unclear...

Was reformulated (see below)

L382: ... potion? (see below)

Was deleted

**Response:** L 380 & L382 The presence of two separate populations of NC10 bacteria at a water depth between 12 and 22 m, in the region where also anaerobic oxidation of methane with denitrification may exist, suggest that this organism was also partially contributing to the anaerobic oxidation of methane with nitrite (*n-damo*). However, again it remains unclear whether *Crenothrix* that also peaked in this region completely reduced dissolved nitrate to $N_2$ or both, NC10 bacteria ($NO_2^-$ reduction) and *Crenothrix* are involved in the N loss processes. In this context it is also worth mentioning that the highest abundance of NC10 bacteria in our

and other studies is often observed at the oxic - anoxic interface (Ettwig et al., 2008) and it is controversially discussed whether *M. oxyfera* can also use external $O_2$ to oxidize methane near the oxycline. Therefore, the respective roles of NC10 and *Crenotrix* remain unclear in this study.

L398: ... as shown...

Was accepted

---

## Author Comment (AC2) · 15 May 2020

We like to express our gratitude for the detailed feedback from Reviewer #2. Below, we have provided a detailed point-by-point list of answers and replies to the comments and suggestions raised by the reviewers. We have had every attempt to address all suggestions and the numerous highly valuable recommendations where appropriate, and have provided detailed responses and explanations below.

Response to general comments:

Reviewer #1 mentioned that the authors seem well aware of the limitations of their isotope results and temper their conclusions with an appropriate amount of the limitations of the presented data (with very few exceptions where a slight over-reach of data interpretation can be identified). In contrast, Reviewer #2 asked for a more cautious interpretation of the data.

Therefore, we have settled in the revised manuscript on a compromise approach.

To address this comment, we acknowledge in the revised version of the manuscript on line 392 the limitation of isotope and microbial community data when activity indicators, for example derived from NanoSIMS analyses, are missing (for example: L338-339: … where *AOM* may affect microbial nitrate reduction, although more canonical heterotrophy could also occur). We also agree with reviewer #2 that microcosm and incubation experiments are excellent approaches to evaluate which processes govern isotope profiles similar to those observed in our study or to determine degradation rates. We are in fact pursuing such experiments (e.g. Kuloyo, 2020). However, it is also known that isotopic fractionation factors especially for transformation processes in the nitrogen cycle observed in laboratory are not always transferrable to field sites, and hence are a useful complement but not a replacement of field studies. Similar, incubation experiments would also show a potential for processes (rates) that could be occurring *in situ*.

Reviewer #2 stated that the paper is prepared with a "certain degree of carelessness with a lot of typos and word adding", but marked only a few typos within the manuscript. We have addressed all typos and stylistic improvements recommended by Reviewer #1 and #2. In addition, we have two native speakers as co-authors who have carefully edited the revised manuscript, to eliminate any remaining spelling and grammatical mistakes, in order to address the concerns of Reviewer #2.

Reviewer #2 expressed some concerns with the modelling portion of the manuscript. The modelling component in the original manuscript constituted only a minor part of our study and was rather used to build up the hypothesis. We agree that the application of a rather simple model demands a more cautious interpretation of the modelling results (micro-aerobic methane oxidation) and we have modified the revised manuscript accordingly. For instance, the conclusion that micro-aerobic methane oxidation will only occurring to a very limited extent and must be verified with a numerical model that will be published elsewhere (see point-by-point answers)

In this context Reviewer #2 asked whether "steady-state conditions" can be assumed and mentioned that our modelled concentrations will be way off by using Kz of 0.1 to 2.1 m$^2$/d. We now explicitly state in the Method part of the revised manuscript that the studied ecosystem is a hydraulically static system, where we assume that advection or mixing do not occur. However, diffusion has to be modelled dynamically, in order to reflect system dynamics adequately and "steady-state conditions" cannot be assumed, as now stated in the revised manuscript.

Reviewer #2 also commented on the use of Kz from literature data. This issue is addressed in detailed in the point-by-point responses. In short, the suggested Kz value by the reviewer of 0.004 cm$^2$/s (0.03 m$^2$/d) is a factor of ten too small compared to (typical) literature data (Oswald, 2015). If we calculate the Kz that is valid for the investigated lake (as suggested by Rev. #2) than the new $K_z$ value fits perfectly to literature data (0.1 m$^2$/d), and the value that was used in the original draft of our manuscript (see also point-by point-answer). As also observed by Reviewer #2, we agree that field data and modelling results do not fit, and hence this supports our conclusion that only diffusion cannot describe the depth-profile of methane concentrations.

Reviewer #2 also commented that the statement in the abstract "that it is an exaggeration to state that nitrate-dependent methane oxidation has the potential …. is not really convincing", while stating that "it does not happen; we know that". Unfortunately, no references were provided that would conclusively document that this process is not occurring at our study site. There are a few studies demonstrating by genome analysis and anaerobic experiments with enrichment cultures that Crenothrix and other microbes can reduce nitrate with methane to N$_2$O (Oswald et al. 2017, Mustakhimov et al. 2013, Naqvi et al. 2018 etc). Therefore, it is worthwhile to discuss the potential of this process to reduce nitrate loading in aquatic environments. These previous publications suggested that this newly discovered process (AOM with nitrate linked with anammox) could have environmental relevance. In this regard, we believe our study and new data provide a valuable contribution to the literature on this topic, while our wording with "has the potential" is very cautious in this context.

Below we provide a point by point response to the comments of the tow reviewers below. Our responses appear in regular font while the original comments by the reviewers appear in italic font.

*Comments of Reviewer #2*

Some of the comments of Reviewer #2 were already discussed at the beginning of our point-by-point answers.

Main points:

*Abstract: In light of much more important N-loss reactions (denitrification anammox) I think it is an exaggeration to state that nitrate-dependent methane oxidation has the potential to reduce nitrate loading. It just does not happen, we know that. I doubt that AOM-denitrification-anammox process really is an overlooked process...it simply is less important than canonical denitrification and aerobic methane oxidation.*

**Response**:

While reviewer #2 states that "it does not happen; we know that", no references were provided that would conclusively document that this process is not occurring at our study site. There are a few studies demonstrating by genome analysis and anaerobic experiments with enrichment cultures that Crenothrix and other microbes can reduce nitrate with methane to $N_2O$ (Oswald et al. 2017, Mustakhimov et al. 2013, Naqvi et al. 2018 etc). Therefore, it is worthwhile to discuss the potential of this process to reduce nitrate loading in aquatic environments. These previous publications suggested that this newly discovered process (AOM with nitrate linked with anammox) could have environmental relevance. In this regard, we believe our study and new data provide a valuable contribution to the literature on this topic, while our wording with "has the potential" is very cautious in this context.

*There is no information on the site/lake. The name and location of the lake should at least be mentioned*

**Response**: In the revised version of the manuscript, the name of the lake has been added to the abstract and we have re-written the first sentence as follows:

L 18: Here, we report vertical concentration profiles and corresponding stable isotope compositions of $CH_4$, $NO_3^-$, $NO_2^-$ and $NH_4^+$ in the water column of Lake Fohnsee, a stratified lake located in southern Germany, which…

*L56: Why is nitrate reduction more important in lakes than nitrite reduction, just because there is more nitrate than nitrite? Nitrite is an intermediate and assuming that the most important N loss pathway is complete denitrification, nitrite reduction has to balance nitrate reduction, if nitrite does not accumulate.*

**Response**: In case that there are no alternative electron donors such as DOC or Fe(II) then nitrite will not be formed and one can assume that nitrate reduction with AOM may be more important.

For clarification we change this sentence and added to L 56: If there are no alternative electron donors such as DOC or Fe(II) available microbial nitrate reduction with AOM may be more relevant in aquatic environments than canonical heterotrophy.

*L88: "...coupled the diffusion model with a degradation term to clarify the effect of dissolved oxygen on methane oxidation. The observed coupled process has the potential to constitute an important sink of dissolved nitrogen (NO3-, NO2-, NH4+) and methane(CH4) in freshwater environments." What exactly is coupled? What coupled process are the authors referring to? This is not clear at this point of the article what they did in the model and how O2 thresholds are integrated. Even if an explanation will follow in the method section, this needs to be clarified (or moved to the more detailed sectionson the model parametrization).*

*Model: There is not enough explanation of the model. Obviously, it is not a realreaction-diffusion model, but it also is not just a diffusion model, right? What are thereaction parameters, how are they set? I am not an expert in modelling, but it remainsunclear how the modelling works, apparently, a purely diffusive part and a reaction partis combined, but the coupling of the model components is unclear. Most importantly,how well constrained is turbulent diffusion? The results (modelled concentration pro-files and isotope ratios in water column) will be highly sensitive to the choice of the D,and adopting values for D (by the way D is used usually for molecular diffusion only) from other lakes may not be appropriate. In fact, the authors seem to have a very limited knowledge of modelling turbulent diffusion in lakes. Firstly, it seems that their choice of what they call D (or K in the literature) is at least two orders of magnitude higher than would be expected for a stratified lake. They cite Oswald et al. from which D was adopted. But looking into the paper by Oswald, I saw that their choice of Kz was 4x10-3 cm2 s-1, which corresponds to approx. 0.035 m2 d-1. If the authors really used D/Kz values between 0.1 and 2.1, their modelled concentrations will be way off. Finally, assuming different turbulent diffusion coefficients for O2 and CH4 is non-sense. Turbulent diffusion is not solute-specific (in contrast to molecular diffusion), itis a hydrodynamic property of the flow field. As for the first-order methane oxidation rate coefficient, how can the authors just assume a value adopted from other studies? This parameter will change significantly between ecosystems, and has to be estimatedbased on fitting of the model to the observational data.*

**Response**: We have re-written this part of the Method section and have already given our view to his comment at the very beginning of the point-by-point answers.

In addition, we have written Kz instead of D, have calculated $K_z$ for Lake Fohnsee and have clarified that the newly calculated Kz fits with literature data. We have already mentioned above that the reviewer's data from literature are not appropriate for this case study.

In the revised text we will use the 1D diffusion model with a degradation term to find some evidence that the observed concentration profile of methane cannot only explained by diffusion. In addition, we have fitted the theoretical methane concentration curve (modelled with diffusion only) to the field data using the rate constant k and compared this value with literature data.

New: Methods

For the 1-D diffusion model, a semi-infinite system was assumed where the lower boundary (at x = 0) is kept at a constant input concentration $C_0$, and the initial concentration throughout the system is zero. The following formula (Eq. 1) from Crank (1975) that represents an analytical solution, which was used to determine the methane concentration as a function of depth (resolved in 0.1 m intervals) along the 10 m long water column below the oxycline at time t:

$$C = C_0\ erfc\ x = \frac{\mathrm{X}}{2\sqrt{(K_z \mathrm{t})}} \qquad\qquad\qquad (\text{Eq. 1})$$

where C [mg/L] is the methane concentration in the water column as a function of distance (depth) x and time, $C_0$ [mg/L] is the constant concentration of methane at the lower boundary, located at a depth of 22 m below the lake surface (bottom of the water column), $K_z$ [m² day$^{-1}$] and represents the turbulent diffusion coefficient for methane in water. For modeling, time t was set to 90 days. This corresponds to the period where stagnant conditions for lake water are assumed to prevail (no mixing and advection) so that methane is transported within the water column by diffusion, only. For methane a turbulent diffusion coefficient of Kz = 1.2*10-6 m2/s, corresponding to 0.1 m2/day, was calculated for Lake Fohnsee according to Wenk et al. (2013) and Bless et al. (2014). This value is at the lower range typically applied for methane flux calculations and modeling (0.1-2.1 m2/day) at stratified lakes such as at Lake Rotsee and Lake Lugano (Oswald et al., 2015; Wenk et al., 2014).

If the diffusing substance is microbial degraded or immobilized, the differential equation for diffusion needs to be extended by additional reaction terms. If first-order degradation is considered, an analytical solution is also available from Crank, (1975), which was used for 1-D modelling of methane diffusion and degradation ( Eq. 2):

$$C = \frac{C_0}{2} \exp\left(-x\sqrt{k/DK_z}\right) \operatorname{erfc}\left(\frac{x}{2\sqrt{K_z t}} - \sqrt{kt}\right) + \frac{C_0}{2} \exp\left(x\sqrt{k/K_z}\right) \operatorname{erfc}\left(\frac{x}{2\sqrt{K_z t}} + \sqrt{kt}\right) \qquad \text{(Eq. 2)}$$

where k is the first-order degradation rate constant [day-1]. Here we used the k-value as fitting parameter and compared it to literature data from Blees et al. (2014) and Roland et al. (2016). If the argument kt in Eq. (2) is large enough so that erfc is approaching 2 at the left hand side and 0 at the right hand side, Eq. (3) simplifies as follows (Crank, 1975):

$$C = C_0 \exp\left(-x\sqrt{k/DK_z}\right) \qquad \text{(Eq. 3)}$$

*Nitrate/nitrite isotope measurements: The authors write: Nitrogen and oxygen isotope ratios of nitrate were calculated by measuring nitrite alone as well as the mixture of nitrite and nitrate in a sample and using an inverse mixing calculation to determine the isotopic ratios of nitrate alone. First of all, there seems to be a duplication in this sentence. I think I understand what the authors did. They measured the isotopic composition of nitrite, and then the isotopic composition of the mixture. Based on mass balance calculation, they then calculate the isotopic ratios of nitrate alone. This works for d15N, but does it work for d18O? I am pretty sure that it does not. In a sample that contains nitrite and nitrate, O isotope fractionation during the conversion to N2O is different for nitrite and nitrate. Hence the d18O of the N2O cannot simply be standardized, because the O-isotope offsets will be different for nitrite and nitrate. In other words, the d18O of a NOx sample is probably meaningless, and so will be the calculated d18O nitrate values. The nitrate d18O should have been measured after removal of the nitrite. Could the changes in Dd15N(nitrate-nitrite) be an artifact that is simply the result of this effect and changing nitrite/nitrate concentration ratios?*

**Reponse:**

Regardless of the method used (bacterial reduction, reduction with cadmium or most recently with titanium), the objective is the production of $N_2O$. The method used for this study is the reduction of nitrate to nitrite on an activated cadmium column and then the conversion of nitrite to $N_2O$ with an azide buffer. This method has the advantage of being able to test the conversion yields at each stage. For each step, international standards are used. In very rare cases in the environment, a significant amount of nitrate and nitrite may be present. Our approach is based not on the addition of an additional reagent, which could also create a bias, but on the conversion of nitrite to $N_2O$ and the conversion of the nitrate+nitrite mixture to $N_2O$. Details of the calculations were recently published in Sebilo et al., 2019 Scientific Reports.

The calculation of the isotopic composition is based on the measurement of the isotopic composition of $N_2O$ with an IRMS and the correction between the values obtained for the standards and the values measured by linear regression. For samples obtained from 14, 16, 18 and 20m depth, both nitrite and nitrate are present. However, taking into account the concentration ratios, the amount of nitrite represents at most 10% of the total concentration for the samples except for the 20m sample where the nitrite concentration is around 1 mg/L and the nitrate concentration is around 0.5 mg/L. For this point, taking into account the two molecules and calculating the nitrate $\delta^{18}O$ gives a value of 5.6‰ whereas it was 5.4‰ without correction.

*L188: How was complete outgassing of CH4 assured before headspace analysis ?Was brine/NaOH added? Concentrations were calculated based on Henry's Law, but what about the d13C? Is there an isotope shift between CH4 in the headspace and the CH4 dissolved? If so, was that considered?*

**Response**:

- outgassing was not complete since, since we followed the headspace equilibration technique by EPA (2002);

- following this EPA technique, we did not add either a salt solution or NaOH to the sample solution;

- We assumed negligible C isotope fractionation between dissolved methane and methane in the headspace (e.g. Feux 1980) and therefore report the measured ☐13C values for headspace methane.

*Results: I am a bit disappointed by the low number of data points/analyses. As a consequence, isotope gradients are not well resolved (and their interpretation is hence complicated), and the profiles are not replicated for several time points. Do the authors assume steady state conditions? How relevant is this for the model fitting?*

**Response:** We agree that a depth resolution of 0.5 m or even lower would have been better, but we sampled up to 3L of lake water for each depth (water samples for IC, isotopes, DOC and microbiology) and we wanted to exclude mixing between the different depths by sampling.

*Figure 3c is very difficult to read? Why not showing profiles (connected symbols) for the most relevant OTUs. It is almost impossible to see the vertical structure.*

**Response:** We now present Figure 3c with the symbols connected for the most relevant OTUs to make it easier to see the vertical structure.

*Discussion: It is not clear to me what the arguments are that allow the authors to exclude oxic methane oxidation. I agree that the concentration profile suggests reaction below the redoxcline, but you do not need to model this to come to this conclusion. At the same time, do the authors assume steady state? Apparently, the lake undergoes seasonal fluctuations, so that the curvature of the concentration profiles may represent a non-steady state, and its interpretation with regards to where reaction takes place and where not is biased.*

**Response**: We agree that a more complex model is needed to perform flux estimation. In addition, it will make sense as suggested by Reviewer #2 to incorporate the isotopes of methane in the modelling part, but this was out of scope for this paper.

Now in section 4.1.

To test the hypothesis if methane diffusion from the lake sediments towards the oxycline can explain the methane concentration profile in the NMTZ, a simple 1D diffusion model with a constant methane input ($C_0$= 2.6 mg/L) was used (Fig. 4).

Our results provide some evidence that diffusion controlled methane fluxes from the sediment surface to the oxycline are highly insufficient for explaining the non-linear decrease of methane concentrations in the water column (Fig. 4). To test the hypothesis whether AOM controls denitrification, a model run with a 1D diffusion model linked with a first-order degradation rate constant, that was used as fitting parameter (Eq. 3), was performed. Results suggest that a k value of 0.09 [d-1] fits reasonable well the observed depth profile of methane. This k-value is in excellent agreement with the results of Roland et al. (2016) for AOM with nitrate from a temperate lake and may support our hypothesis that AOM with nitrate affects the observed concentration profiles of methane and nitrate. However, micro-aerobic methane oxidation can also play an important role in the water column of this lake.

*Can the authors explain why a 90% decrease in ammonium is associated with aïĄd'15N shift of only 4‰*

*L335-7: The authors say that above 20 m water depth, there is no evidence for ammonium oxidation. Why? Because the d15NH4 values do not increase? But they also do not increase much below that depth, where the authors suggest that anammox occurs. And most strikingly, the ammonium profile is essentially linear all the way up to the oxycline. To me this suggests that not much ammonium oxidation is taking place at this depth, and essentiall all NH4 is oxidized at the oxycline.*

**Response**: We do not understand that Reviewer #2 mentioned a 90% decrease in ammonium concentration that is linked to a small N isotope fractionation in ammonium. We have observed a decrease of ammonium from 1 mg/L at the bottom of the lake to 0.8 mg/L at a depth of 20 m and simultaneously an increase of $\delta^{15}N$ of ammonium of 4 ‰. Above this depth $\delta^{15}N$ values of ammonium remain constant. Above a depth of 20 m, ammonium concentration decreases from 0.8 mg to below detection at a water depth of 12 m, probably by diffusion/ and maybe by diffusion controlled aerobic NH4-oxidation etc. as outlined in our manuscript (L 335) and, therefore, no significant isotope fractionation is expected and was observed.

In addition, Wunderlich et al. (GCA 2018) has suggested a transport limitation model, where such small stable isotope fractionation of 4‰ can be explained and also Wenk et al. (2013, Limnol. & Oceagr.) found small stable isotope enrichment of 8 ‰ at Lake Lugano for anammox (here NH4-concentrations decreased below detection). We will add both references to the revised manuscript and will also give an explanation based on the paper of Wunderlich et al. (2018). To use the expected form of the concentration line when degradation occurs, as suggested by the reviewer is risky because of the low number of data points.

New text § 4.2:

We also present several lines of qualitative and quantitative evidence that anammox has occurred with AOM coupled with denitrification at the bottom of the NMTZ of the lake. As expected the nitrite concentration at a water depth of 20 m was highest where nitrate reduction occurred (Fig. 2A). Between this depth and the lake bottom, our data strongly suggest that anammox is the main sink of $NH_4^+$. Ammonium occurs in concentrations of up to 1 mg/L at the bottom of the water column at 22m, likely stemming from the heterotrophic degradation of organic nitrogen (e.g., proteins and amino acids) close to the sediment – water interface, and is subsequently transported from the methane zone near the lake sediments into the overlying water column (Norði et al., 2013; Wenk et al., 2014), where the $NH_4^+$ concentration decreases continually towards $< 0.15$ mg/L at 12 m depth. The decrease in ammonium concentration with increasing water depth from around 1 mg/L to 0.8 mg/L is accompanied by an enrichment of $^{15}N$ in the remaining ammonium shifting the $\delta^{15}N$-NH4 values from 7.9‰ to 11.6‰ between 22 and 20 m water depth (Fig. 2C), suggesting that ammonium is oxidized anaerobically while enriching the remaining substrate in $^{15}N$. Above this water depth there is no isotopic evidence that ammonium is oxidized under anaerobic conditions and the decrease of ammonium concentrations may be only affected by diffusion and by ammonification and nitrification processes that may occur at the oxycline. To explain the moderate isotopic shift of 4 ‰ in $\delta^{15}N$-NH4 of ammonium Wunderlich et al. (GCA, 2018) has suggested a transport limitation model, where such small isotope fractionation can be explained and also Wenk et al. (2013, Limnol. & Oceagr.) found a small stable isotope enrichment of around 8 ‰ at Lake Lugano for anammox when almost all ammonium was oxidized.

*340-345: The authors cite the anammox isotope effect study by Brunner and colleagues. But they mix up equilibrium and kinetic N isotope effects between nitrite and nitrate. The inverse kinetic N isotope effect, which applies to active nitrate production from nitrite by anammox, is much lower than the -61‰ mentioned.*

**Response:** We agree and will delete this reference.

*The authors should explain better why anammox could produce a d18O vs d15N NO3 relationship of 0.5. Is this slope consistent with nitrate production from nitrite with the incorporation of O atoms from water? Such slopes in d18O vs d15N NO3 plots have been observed in several ground/freshwater studies. Does this imply that in all these environments anammox was the main N-loss pathway?*

**Response:** Please note that we consider a system that may be controlled by anaerobic redox conditions. This was also clearly stated in the manuscript (… to demonstrate that in a $d^{18}O$ vs. $d^{15}N$ plot for nitrate a slope lower than 1 is a powerful indicator for the occurrence of *anammox* in an anoxic environment). Slopes lower than 1 can also be observed at aerobic/ anaerobic interfaces such as groundwater systems, the oxycline of stratified lakes or as outlined in Wunderlich et al. (2018 in GCA) by specific organisms that reoxidize nitrite to nitrate and do not imply that anammox is the main N-loss pathway in all environments. As a result, we developed several lines of evidence to come to the conclusion that anammox may have occurred.

The comment whether the slope is consistent with nitrate production from nitrite with the incorporation of O atoms is somehow exaggerated because we think that the reviewer knows that this depends on the enzymes involved etc. Here we relegate to the paper of Granger and Wankel (2016) in PNAS.

*What is the relative abundance of "normal" nitrate and nitrite reducers compared to NC10 and Crenothrix?*

**Response**: Because nitrate and nitrite reduction is such a widespread trait held by many facultative anaerobic bacteria, it is not possible to use our 16S rRNA gene sequence data to specifically show the abundance of 'normal nitrate and nitrite reducers' as the reviewer requests. However, the Gamma proteobacteria are very abundant in our samples, and are well known to have many species that are capable of nitrate and nitrite reduction, a trait that is widespread throughout this class. Since the Gamma proteobacteria relative abundance increases with depth into the anoxic zone (Fig 3b), it is likely that many of the Gamma proteobacteria in deeper waters of the lake are responsible for nitrate and nitrite reduction, and denitrification. We will add an explanation to the revised manuscript."

Minor points:

These few suggestions were accepted and improvements were made in the revised manuscript.

References:

KULOYO, O., RUFF, S. E., CAHILL, A., CONNORS, L., ZORZ, J. K., HRABE DE ANGELIS, I., NIGHTINGALE, M., MAYER, B. & STROUS, M. (2020): Methane oxidation and methylotroph population dynamics in groundwater mesocosms. Environmental Microbiology, 22(4): 1222-1237 (early access February 2020; published April 2020). https://doi.org:10.1111/1462-2920.14929

SEBILO, M., ALOISI, G., LAVERMAN, A. M., MAYER, B., PERRIN, E., VAURY, V., MOTHET, A. & LAVERMAN, A. M. (2019): Controls on the isotopic composition of nitrite ($\delta^{15}$N and $\delta^{18}$O) during denitrification in freshwater sediments. - Scientific Reports, 9: 19206. Doi.org/10.1038/s41598-019-54014-3; published December 16, 2019.

WUNDERLICH, A., HEIPIEPER, H.J., ELSNER, M. AND EINSIEDL, F. (2018) Solvent stress-induced changes in membrane fatty acid composition of denitrifying bacteria reduce the extent of nitrogen stable isotope fractionation during denitrification. Geochimica et Cosmochimica Acta 239, 275-283.

---

## Author Response (AR2)

Comments on

**Biogeochemical evidence of anaerobic methane oxidation and anaerobic ammonium oxidation in a stratified lake using stable isotopes" by Florian Einsiedl et al.**

We like to express our deep gratitude for the constructive suggestions made by Reviewer #1.

We agree with Reviewer #1 that a simple mass balance approach cannot be used for disentangling analyses of oxygen isotopes of nitrate and nitrite combinations without nitrite reference materials. Therefore, we have followed the suggestion of Reviewer #1 and have removed the O isotope data of nitrite in the revised manuscript, especially since they provided limited usefulness for arriving at the major conclusions of this manuscript.

Removal of the oxygen isotope data of nitrite required the following changes in the revised manuscript:

Line 104 now reads: Samples for isotope analysis of nitrite ($\delta^{15}N$), nitrate ($\delta^{15}N$, $\delta^{18}O$) ...

Line 166: $\delta^{15}N$ and $\delta^{18}O$ values of nitrate and $\delta^{15}N$ values of nitrite and ammonium were obtained

Line 189: The precision for $\delta^{15}N$ values of nitrate and nitrite was $\pm$ 0.5‰ and for $\delta^{18}O$ of nitrate $\pm$ 0.8‰.

Line 256: Simultaneously, $\delta^{15}N$ of nitrite increased from 0.1‰ to 18.7‰ concurrently with increasing $\delta^{15}N$ values of nitrate (Fig. 2C).

From line 256 onwards, the following text was deleted:

The $\delta^{18}O$ values of nitrite were near -5‰. According to Casciotti et al. (2007), the here measured values are 9‰ lower than expected according to a situation where $\delta^{18}O$ values of nitrite was established in equilibrium with lake water with a $\delta^{18}O$ of -10‰. However, Sebilio et al. (2019) found that the $\delta^{18}O$ values of nitrite are lower (< +4‰), as observed in our study, when the oxygen exchange reaction is controlled by biotic exchange processes compared to those suggested by Casciotti et al. (2007), who studied abiotic exchange reactions between nitrite and water-oxygen.

We also like to express our gratitude for the feedback from Reviewer #2. Below, we have provided a point-by-point list of responses to the comments and suggestions raised by the reviewer.

*I am wondering about the fact that there was not an appropriate d18O_NO2 standard used.*

**Response**: We agree and have deleted the $\delta^{18}O$ values of nitrite from the revised manuscript following the suggestion of both reviewers.

*I am still not convinced by the way they sell their data set (evidence for true AOM is weak), and I still think there are some issues, but I guess that this is part of the scientific exchange. We do not need to all agree. It is definitely a nice data set, certainly worth to be published.*

**Response**: We agree with reviewer 2 that a more refined depth resolution of the samples and the subsequently obtained data would have been beneficial for further strengthening the evidence for AOM; as pointed our previously in the manuscript, this was however not possible and therefore we have very carefully worded this section by stating "some evidence …." in lines 294, 326, and 333

*The authors tone down in their response the value of incubation experiments. I do not agree. Experiments, even if not directly reflecting the natural conditions, would have helped to calibrate the observed isotopic signatures, and to gain confidence in their interpretation. I was not referring to isotope effect experiments, and I am aware that rate measurements will deliver potential rates only, but clearly, rate measurements would have helped to indicate the potential for the use of specific substrates, and thus would have helped to support (or not) the claimed links between different biogeochemical reactions (anammox and AOM).*

**Response**: We thank the reviewer for this clarification. Fact is that we have not conducted any complementary laboratory experiments but we agree that laboratory experiments could help to study the potential of specific substrates for microbial turnover.

*The authors state that there is barely any mixing during summer ("no mixing, no advection"); I assume they mean no advective mixing, because turbulent diffusive mixing, even if sluggish will take place.*

**Response**: This is an excellent point and we have modified the manuscript accordingly. Now line 134 reads as follows: This corresponds to the period where stagnant conditions for lake water are assumed to prevail (no advective mixing)…..

*As for the choice of a value for Kz: In my first review; I was not suggesting a value of 0.03 m2/d. I was referring to a paper by Oswald et al. from Lake La Cruz (Oswald et al. 2016), for which this value was reported. I mixed up the different studies/lakes investigated by Oswald et al., and values from Lake Rotsee, as well as values for Lake Lugano seem to be higher, indeed. But the broad range of Kz values observed for the different meromictic lakes highlights that there is not "a typical literature value", and that just assuming a value for Kz is difficult.*

**Response**: Thank you for this clarification.

*As for the interpretation of the NH4 concentration versus the d15NH4 profiles: I see a more or less steady decrease from 1 to <0.1 mgN/l between 22 and 12 m, without a strong overall 15N enrichment. From these patterns, I find it quite difficult to pinpoint anaerobic ammonium oxidation in the deep water column. Within the error of the d15NH4 analyses, the isotope profile looks almost straight to me….an observation that could most plausibly be explained by aerobic ammonium*

*explanation at the redox transition zone, which serves as an efficient NH4 sink without much N isotope fractionation.*

**Response**: We clearly stated from line 342 to 344 where ammonium concentrations decreased and $\delta^{15}N_{NH4}$ increased:

The decrease in ammonium concentration with decreasing water depth is accompanied by an enrichment of $^{15}N$ in the remaining ammonium shifting the $\delta^{15}N_{NH4}$ values from 7.9‰ to 11.6‰ between 22 and 20 m water depth (Fig. 2C), suggesting that ammonium is oxidized anaerobically while enriching the remaining substrate in $^{15}N$.

We also like to point out that we did not interpret the N isotope ratios and concentrations of ammonium in isolation, but considered the following observations in combination:

- both nitrite and ammonium concentration trends;
- $\Delta\delta^{15}N$ between nitrate and nitrite increased from 11‰ in NMTZ to > 26‰ at the water depth of 20 m, where $\delta^{15}N$ values of ammonium increased while $NH_4^+$ concentrations decreased (Fig. 2C);
- the deviation of the slope of $\delta^{18}O$ versus $\delta^{15}N$ values on a dual isotope plot (2D plot) for nitrate from the expected value of 1 for microbial denitrification is a powerful tool to identify anammox.
- This, together with the presence of '*Candidatus Anammoximicrobium*' within the strictly anaerobic water column, is in our view sufficient to come to the conclusions that these combined observations may indicate that anammox has occurred at Lake Fohnsee between a water depth of 20-21 m.

Reviewer 1 also expressed the opinion that the statements about the key findings are reasonably tempered in the revised manuscript.

*I am still confused with regards to the d18O_NOx values. Again, in a sample that contains nitrite and nitrate, O isotope fractionation during the conversion to N2O (independent of the method of conversion) must be different for nitrite and nitrate (via nitrite), simply because one versus two O atoms will be plucked off during the fractionating transformation, respectively. Hence it remains unclear to me how the d18O of the combined NO3+NO2 sample is standardized. The problem is not the yield, the problem is that the N2O from the nitrate will likely have a much higher d18O than the d18O of the N2O generated from the NO2.*

**Response**: As mentioned in our response to Reviewer #1, we agree with the major limitation of the $\delta^{18}O$ values of nitrite as no reference material was used, which also impacts mixing calculations. However, for most of the nitrate samples (14m to18m water depth) nitrite represents at most 15% of the total nitrate+nitrite concentration. Recalculations using a value of +4‰ for $\delta^{18}O$ of nitrite revealed that the $\delta^{18}O$ values of nitrate are between 2 and 3‰ lower compared to those calculated with the previously reported $\delta^{18}O$ values of nitrite (ca. -4‰). These changes had no consequences for our conclusions reported in the earlier version of the manuscript.

In the revised manuscript we have re-calculated the $\delta^{18}O$ values of nitrate.

Now from line 175 the revised manuscript reads: For $\delta^{18}O$ values of nitrate we performed a mass-weighted isotope mass balance calculation assuming that at a pH of 7 the $\delta^{18}O$ of nitrite is in equilibrium with water with a value close to +4‰.

Figs 2C and 5 were corrected using the new data set of $\delta^{18}O$ of nitrate and additional information was given for Fig. 2C.

*L915: established is misspelled*

**Response:** This part was deleted

In summary, we have made every attempt to address the suggestions and the highly valuable recommendations by both reviewers, and have provided detailed responses and explanations above. We hope that the revised manuscript can now be accepted for publication.

[revised manuscript text omitted]

**Kommentiert [EF7]:** Delta18O of nitrite was deleted from Fig. 2C

[Figure]

**A.**

Percent of total 16S rRNA gene sequences

**ANOSIM**
R: 0.57
*P* = 0.002

Geochemical zones
- ■ Oxic surface
- ○ NMTZ top
- ● NMTZ bottom
- ▲ Methane zone

**B.**

Percent of total 16S rRNA gene sequences

From left to right in histograms

**C.**

16S rRNA gene copies mL

Water depth (m)

oxic - anoxic interface

- + Total 16S rRNA gene copies
- ● *Crenothrix* OTU_1
- ■ *Crenothrix* OTU_2
- ● NC10 group OTU_1
- ■ NC10 group OTU_2
- ● 'Candidatus Anammoximicrobium' OTU_1

qPCR normalized
abundances of
16S sequences

**Figure 3A-C. Analysis of 16S rRNA gene data from microbial communities in the stratified lake. (A) Heatmap showing the relative abundance of specific groups in the 16S rRNA gene sequencing data, and corresponding heirarchical clustering analysis (analysis of similarity (ANOSIM) P value = 0.002) of four geochemically defined zones.  For those depths where replicates were obtained, the data for both replicates are shown.**

**(B) The relative abundance of 16S rRNA gene sequences affiliated with the major groups across the stratified water column.  (C) Abundance of 16S rRNA gene copies determined via qPCR, and the qPCR normalized absolute abundances of 16S rRNA gene sequence relative abundances from key populations (OTUs) potentially involved in AOM and anammox, specifically those affiliated with Crenothrix, NC10, and potential anammox bacteria.**

[Figure]

**Figure 4.** Depth-profiles of methane concentration (filled triangles) and its isotopic composition (filled circles) within the water column, modelled methane concentrations (open triangles) using a 1D-diffusion model with a turbulent diffusion coefficient for $K_{zCH4}$ of 0.1 $m^2$ $day^{-1}$ (model diffusion), and a 1D diffusion model additionally linked with a degradation term (first order rate constant k= 0.03 $d^{-1}$ (model diffusion and reaction).

[Figure]

**Figure 5. δ¹⁸O versus δ¹⁵N plot of nitrate with the typical trajectory of 1 for denitrification obtained under laboratory conditions (black line), calculated trajectory of 0.65 for the *n-damo* zone (20m and above, blue line) and around 0.5 for the *anammox* zone (20-22m, green line).**

[Figure]

**Figure 6. Conceptual model of the coupled N and C cycles in the anoxic water column of Lake Fohnsee**

[Figure]

Copernicus Publications
The Innovative Open Access Publisher

---

## Author Response (AR3)

Reviewer #1

We like to express our deep gratitude for the detailed and constructive feedback from two reviewers. Below, we have provided a detailed point-by-point list of answers and replies to the comments and suggestions raised by Reviewer #1. We have made every attempt to address the excellent suggestions and the numerous highly valuable recommendations where appropriate, and have provided detailed responses and explanations below.

Reviewer #1 agrees with our assertion, that links between nitrate-AOM and anammox have not been widely demonstrated in the literature, and that our study is an important step in developing an environmental understanding of these processes. Reviewer #1 found the study well executed and the data of high quality.

Response to general statement:

Reviewer #1 mentioned that the authors seem well aware of the limitations of their isotope results and temper their conclusions with an appropriate acknowledgement of the limitations of the presented data (with very few exceptions where a slight over-reach of data interpretation can be identified). In contrast, Reviewer #2 asked for a more cautious interpretation of the data.

We have settled in the revised manuscript on a compromise approach that is partially based on opinion of Reviewer #1 that most of our original conclusions were well tempered, while we have also made several text additions in the revised manuscript that further caution against an over-interpretation of our findings (L 419, and Ls 330, 362, 426).

Point-by-point response:

**Point-by-point response:**

*Comments of Reviewer #1*

*The quality of the figures seems sub-par and some effort should be given to improve*

*on details such as text and symbol sizes and colors, axes labels and ticks, etc.*

**Response:** 1 & 2: We agree that an improvement of the quality of the figures was necessary and we also have changed the expression of concentrations to mmol/L. (Figs. 1-4 were improved as suggested).

*L137: What is the reasoning behind the two diffusion coefficients for methane? This is presented in an apparent attempt to bracket a range of acceptable flux estimates, but is not explained in the text.*

**Response:** The reason was that we have not calculated the site-specific $K_z$ value in our first draft of the manuscript and used the upper and lower ranges of literature data. The modelling was discussed in detail by Reviewer #2 and we made every attempt to address both

recommendations. In the revised manuscript we have calculated the turbulent diffusion coefficient Kz for Lake Fohnsee as suggested by Reviewer #2, and have improved the modelling section by fitting the methane profile only.

*Something about Equation 1 seems incorrect. Are there meant to be two equalities here? Do the 'x' terms both refer to depth? In general, I think 'z' is more frequently used for referencing to depths. Please confirm that the expression of this diffusion equation is properly written*

**Response:** In the revised manuscript, we have corrected the diffusion equation after Clark (1975), have extended the used equations for clarification (see Rev. #2), and define in the manuscript that "z" represents the depth. (from Line 121)

*L310: I understand that in the presence of NO3-, sulfate reduction is not thermodynamically predicted to proceed, however some arguments have been made for processes occurring in micro-zones inside of particles, for example. How much anticipated change in sulfate would be predicted – and would the IC measurements actually be sensitive enough to this? The increasing presence/abundance of Deltaproteobacteria also lend some credence to the idea that at least some level of SO42- reduction could be occurring. Changing units into molarity would help readers with this comparison as I noted above.*

To address this, we revised the manuscript to the following new text:

L 313

**Response:** Sulfate concentrations were clearly above the detection limit of the IC and we observed a decrease of sulfate concentrations from 0.08 mmol/L at a water depth of 21 m to around 0.07 mmol/L close to the lake sediments. This 14% decrease in sulfate concentration with increasing depth could be interpreted to indicate partial bacterial sulfate reduction in micro-environments of particles near the lake sediment surface as suggested by the reviewer or alternately, by mixing effects between sulfate-free water from the sediments, where methanogenesis may occur, and lake water. As we also found measurable nitrate concentrations at the same depth (22 m) where we observed decreasing sulfate concentrations we can only speculate what processes control decreasing sulfate concentrations.

*The δ18O values of nitrite are reported, but nowhere in the text is it explained how these values were determined or calibrated. Further, given the low pH of lake water, the δ18O values of nitrite are very likely to be in isotopic equilibrium with the water, yet appear to fall around -4 to -6‰ which would be much too low. Given a lake water δ18O value of ~ -10‰ – the δ18O value for nitrite in equilibrium should fall closer to +4‰ (see Casciotti et al., 2007). Finally, the δ18O values of nitrite in this study are not mentioned or involved in aspect of the conclusions – and should probably be omitted for clarity (e.g., they aren't used to bring any new insight into the system as presented).*

From line 164 and from line 354

**Response**:

We used international nitrate standard with known isotopic composition ($\delta^{15}$N & $\delta^{18}$O values) and a lab-internal standard for $\delta^{15}$N of nitrite but not for $\delta^{18}$O of nitrite, while using the measurement gas $N_2O$.

The isotopic composition of nitrite was determined using the azide method, similar to the analysis of nitrate. In order to ensure the proper reduction of nitrite to $N_2O$, in addition to the samples, internal laboratory standards for KNO2 were analyzed in each batch (Lb1, $\delta$15N = -63‰ and Lb2, $\delta$15N = +2.7‰). Corrections of the raw $\delta^{15}$N values were made based on the known values of the nitrate and nitrite standards.

With respect to the observed $\delta^{18}$O values of nitrite, the paper by Casciotti et al. (2007) demonstrates perfectly that there is an isotopic exchange between oxygen of the water and oxygen of nitrite. Once this exchange is achieved, an isotopic equilibrium is established depending on the isotopic fractionation. This fractionation leads to significantly higher $\delta$18O values of nitrite compared to those of water. In Casciotti's study, the isotopic fractionation determined for freshwater is around -14‰. So based on the $\delta$18O of the water in this study close to -10‰, the expected $\delta$18O for nitrite should be +4‰, assuming there is only abiotic exchange. More recently, Sebilo et al. (2019) published a study based on isotope tracing during nitrite or nitrate reduction. This study revealed that the oxygen isotope shift was immediate and the authors attribute it primarily to denitrifying bacteria, given the rapidity of exchange. In this study, the $\delta$18O of the water was close to -10‰ and the $\delta$18O of the nitrite during its reduction was relatively constant, oscillating between 0 and -2‰, and hence displaying lower values than those expected with abiotic exchange alone

The results obtained in the here discussed manuscript, with relatively constant $\delta$18O values for nitrite close to -5‰ indicate that an isotopic exchange occurred between the oxygen of the water and the oxygen of nitrite and that the latter was predominately controlled by biotic reactions. Moreover, since denitrification alone should have resulted in a $\delta^{18}$O value of the nitrite between -2 and 0‰, this discrepancy may suggest that another biotic process is taking place or the extent of isotope fractionation depends on the microorganisms controlling this process.

**Minor revisions:**

L15: "*Nitrate dependent anaerobic methane oxidation and anaerobic oxidation of ammonium (anammox) have the potential...*"

**Response:** Here it is not clear what changes the reviewer would like to see. We suggest to change the sentence as follows:

The nitrite (n-damo)/ nitrate dependent anaerobic methane oxidation and the anaerobic oxidation of ammonium (anammox) represent two microbially-mediated processes that can reduce nitrogen loading of aquatic ecosystems and associated methane emissions to the atmosphere. All other minor revisions focusing more or less on awkward wording were accepted and have greatly improved the manuscript that now reads as follows:

L20: *anammox does not require italics.*

**Response**: was changed throughout the manuscript

L24: ….*most likely explanation…*

**Response**: …is the most parsimonious explanation

L30: *… that consist of bacteria known to be involved in…*

**Response**: The associated methane concentration and stable isotope profiles indicate that some of the denitrification may be coupled to AOM, an observation supported by an increased concentration of bacteria known to be involved in n-damo/ denitrification with AOM (NC10 and Crenothrix) and anammox ('Candidatus Anammoximicrobium')….

 L46: *…coupled to nitrate…*

**Response**: misspelling was improved

L54: *…,*whereas ANME-2d lineage uses methane to reduce nitrate to nitrite (Raghoebarsing et al., 2006).

L56: *Beside*

**Response**: was deleted

L59: *…related to Crenothrix also have the…*

**Response**: …. related to *Crenothrix* also have

L56: *Crenothrix may likely act as driver*

**Response**: … Crenothrix may act as a driver…

L70: *references do not need to be italicized* "was corrected"

**Response**: (Shen et al., 2014; Zhu et al., 2018) was improved

L86: *sequencing of 16S rRNA genes that provides effidence for the…*

**Response**: Our findings show that microbially mediated linkages between n-damo/ denitrification with AOM and anammox have the potential to constitute an important sink of both dissolved nitrogen (NO3-, NO2-, NH4+), and methane (CH4), in stratified freshwater ecosystems.

L109: *… sterile filter, which was then kept frozen…*

**Response**: … The filter including the microbial biomass was kept frozen at

*L149: … which represents the lower bound…*

**Response:** Here we have re-written this § and detailed answers can be found in our response to Reviewer #2 and the manuscript **(from line 122)**

 *L150 this should be moved to introduction or discussion.*

**Response**: As a short introduction to the stable isotope section, we are of the opinion that this text fit well in the current section, and hence have made no changes.

**Response:** The suggested improvements of the reviewer were accepted and we made the following changes in the revised manuscript:

L166: ...(2007), respectively.

**Response**: was added

*L166: N$_2$O ...using sodium azide.*

**Response**: Nitrite was converted to N$_2$O using acetic acid buffer sodium azide.

L170: *the mixture of both nitrate and nitrite was reduced to N$_2$O.*

**Response**... mixture of both nitrate and nitrite was reduced to N2O via azide.

L175: *Can you provide some estimate of error propagation for this inverse mixing calculation?*

**Response**: Please also compare answer to Rev #2

The calculation of the isotopic composition is based on the measurement of the isotopic composition of N$_2$O with an IRMS and the correction between the values obtained for the standards and the values measured by linear regression. For samples obtained from 14, 16, 18 and 20m depth, both nitrite and nitrate are present. However, taking into account the concentration ratios, the amount of nitrite represents at most 10% of the total concentration for the samples except for the 20m sample where the nitrite concentration is around 1 mg/L and the nitrate concentration is around 0.5 mg/L. For this point, taking into account the two molecules and calculating the nitrate $\delta^{18}$O gives a value of 5.6‰ whereas it was 5.4‰ without correction.

 L177: ...*an azide buffer for subsequent analysis.*

**Response**:... by buffered azide solution for subsequent analysis.

L181: *How were the δ15N values of the nitrite standards determined – and to what level of precision? There is no mention of calibrated δ18O standards for nitrite. Yet δ18O of nitrite data are reported (albeit not discussed). Please clarify or omit.*

**Response:** We have the revised the text as follows:

"The isotopic composition of nitrite was determined using the azide method, similar to the analysis of nitrate. In order to ensure the proper reduction of nitrite to N$_2$O, in addition to the samples, internal laboratory standards for KNO2 were analyzed in each batch (Lb1, δ15N = -63‰ and Lb2, δ15N = +2.7‰). Corrections of the raw $\delta^{15}$N values were made based on the known values of the nitrate and nitrite standards (Lines 167-176)

L192: *Here it is unclear if the methane isotope analyses were conducted on the same bottles? Viamanualinjection? Wasthisafullbottlepurgeandtrapapproach? Wasthis automated? Were there standards included in this approach? How were the analyses standardized (e.g., extractions of methane of known composition from water?)?*

From Line 190

**Response**: As stated in the original text, "the concentrations and carbon isotope ratios of dissolved methane in the lake water samples were determined using the static headspace equilibrium technique (EPA, 2002) where 10% of the water sample in the capped bottles was replaced with helium followed by outgassing of the dissolved gases in the water sample into the headspace for 1 h at 25ºC.

In the revised text, we have now clarified that:

- methane concentration and C isotope ratios were determined from the same bottle;

- that only 10% of the bottle content was replaced with an inert headspace gas;

- that this process was not automated;

- standardization of the measurements was accomplished as follows: Instrument stability and linearity was ensured by daily measurements of an in-house methane mix of 5% CH4 (balance helium). Carbon isotope analyses of methane were standardized by measurements of Isometric Instruments (Victoria, BC, Canada) gases containing methane with known $\delta$ 13C values including the following:  B-iso1 ($\delta$13C = -54.5‰, $\delta$2H = -266‰), L-iso1 ($\delta$13C = -66.5‰, $\delta$2H = -171‰), and H-iso1 ($\delta$13C = -23.9‰, $\delta$2H = -156‰);

- standard solutions with dissolved methane of know isotopic compositions were not available;

**Response:** The improvements suggested by the reviewer were accepted and we made the following changes in the revised manuscript:

 *L245: Aerobic redox conditions….*

**Response***:* Aerobic conditions...

*L248: The average concentrations of nitrate …*

**Response***: nitrate concentrations decreased...*

*L265: .. too low for stable carbon isotope analyses.*

**Response** ... were too low for stable isotope analyses.

*L298-300 – Rephrase. Awkward wording.*

From Line 295

**Response:** Here we have decided to remove the $O_2$ calculations because the use of an analytical model has to many uncertainties for a solid evaluation of the effect of micro-aerobic methane oxidation within the oxycline. Instead we focus our interpretation of the difference of observed and modelled methane concentration profiles using the calculated

turbulent diffusion coefficient $K_z$ and calculated a first order rate constant k for AOM with nitrate and compared it to literature.

**Response:** The suggested improvements by the reviewer were accepted and we made the following changes in the revised manuscript:

L321: *A stable isotope technique was used...*

**Response**: Stable isotope data was used

L327: *We also present several lines of qualitative and quantitative evidence that anammox has occurred with AOM coupled with denitrification at the bottom of the NMTZ of the lake.*

**Response**: Several lines of qualitative and quantitative evidence indicate the co-occurrence of anammox, denitrification, and AOM towards the bottom of the NMTZ.

L361: *Here the language reads as though AOM coupled to denitrification has been unequivocally demonstrated, which isn't exactly the case. I think here it is best to qualify this a bit more.*

**Repsonse**: We observed a difference of δ15N values (Δδ15N) of nitrate and nitrite of around 11‰ in the NMTZ at depths of 16 and 18 m, where we suggest the microbial linkage of AOM and denitrification maybe via n-damo. But again, it is also possible that some of the denitrification is coupled to heterotrophic nitrate and nitrite reduction

L370: *This strongly suggests that the additional isotopic difference in $\delta^{15}N$ values between nitrate and nitrite of around +15‰ is likely the result of production of highly $^{15}N$ enriched nitrate derived from anammox.*

**Response:** This is consistent with the additional isotopic difference in δ15N values between nitrate and nitrite of around +15‰ arising as the result of production of highly 15N enriched nitrate deriving from anammox (Δδ15N of +31‰).

L375: *...superimposed on 'normal' isotope effects...*

**Response**: was superimposed on "normal" isotope effects

L382: *As written this statement is incorrect.*

*During anammox when nitrite is reduced with ammonium as electron donor to nitrate, one oxygen molecule from water with a $\delta^{18}O$ value of around -10‰ is incorporated into the newly formed nitrate. Additionally, $\delta^{18}O$ values of nitrite are lowered due to rapid oxygen isotope exchange with water oxygen (Buchwald and Casciotti, 2010; Casciotti et al., 2007; Casciotti et al., 2010) and remained, therefore, constant over the water depth of 16 m to 20m*

**Response:** We have added the statement of an additional isotope effect to the sentence.

During anammox, when nitrite is reduced with ammonium as electron donor and nitrate is produced, one oxygen atom from water having a δ18O value of around -10‰ is incorporated into the newly formed nitrate. This incorporation of a new O atom is also most likely associated

with a kinetic isotope effect – as has been demonstrated for nitrite oxidizing bacteria (see Buchwald and Casciotti, 2010) (Fig. 2c). As a result, the anammox process leads to δ18O values of nitrate remaining low, while δ15N of the remaining nitrate is affected by an inverse nitrogen isotope effect and values continue to increase.

L385: *As a result, the anammox process facilitates δ$^{18}$O values of nitrate remain low,*

**Response**: ... the anammox process leads to δ$^{18}$O values of nitrate remaining low, while...

L361: *..by an inverse isotope effect*

Was added... nitrate is affected by an inverse nitrogen isotope effect and values continue to increase.

*L368: Although not mentioned, I am curious whether any nitrite oxidizing bacteria were detected in the genomic analyses? I assume from their omission that they were not. This could be a useful fact to mention if so.*

**Response:** Because nitrate and nitrite reduction is such a widespread trait held by many facultative anaerobic bacteria, it is not possible to use our 16S rRNA gene sequence data to specifically show the abundance of 'normal nitrate and nitrite reducers' as the reviewer suggested. However, the Gammaproteobacteria are very abundant in our samples, and are well known to have many species that are capable of nitrate and nitrite reduction, a trait that is widespread throughout this class. Since the Gammaproteobacteria relative abundance increases with depth into the anoxic zone (Fig. 3b), it is likely that many of the Gammaproteobacteria in deeper waters of the lake are responsible for nitrate and nitrite reduction, and denitrification.

Now Lines 426

L399: *…and may outcompete the nitrate reducing ANME-2d with lower doubling times in the denitrification zone of stratified lakes (Deutzmann et al., 2014).*

**Response**: ... environmental conditions, helping any nitrate reducing ANME-2d (with lower doubling times) in the denitrification zone...

Was accepted

L408: *... the meaning behind this sentence is unclear...*

**Response:** In this context it is also worth mentioning that the highest abundance of NC10 bacteria in our and other studies is often observed at the oxic - anoxic interface (Ettwig et al., 2008) and it is controversially discussed whether M. oxyfera can also use external O2 to oxidize methane near the oxycline. Therefore, the respective roles of NC10 and Crenotrix in nitrite reduction and nitrate reduction, respectively, linked with AOM remains unclear in this study.

L425: *…as sown…*

**Response**: ... as shown...

We agree with Reviewer #1 that a simple mass balance approach cannot be used for disentangling analyses of oxygen isotopes of nitrate and nitrite combinations without nitrite reference materials. Therefore, we have followed the suggestion of Reviewer #1 and have removed the O isotope data of nitrite in the revised manuscript, especially since they provided limited usefulness for arriving at the major conclusions of this manuscript.

Removal of the oxygen isotope data of nitrite required the following changes in the revised manuscript:

Line 104 now reads: Samples for isotope analysis of nitrite ($\delta^{15}N$), nitrate ($\delta^{15}N$, $\delta^{18}O$) ...

Line 166: $\delta^{15}N$ and $\delta^{18}O$ values of nitrate and $\delta^{15}N$ values of nitrite and ammonium were obtained

Line 189: The precision for $\delta^{15}N$ values of nitrate and nitrite was $\pm$ 0.5‰ and for $\delta^{18}O$ of nitrate $\pm$ 0.8‰.

Line 256: Simultaneously, $\delta^{15}N$ of nitrite increased from 0.1‰ to 18.7‰ concurrently with increasing $\delta^{15}N$ values of nitrate (Fig. 2C).

From line 256 onwards, the following text was deleted:

The $\delta^{18}O$ values of nitrite were near -5‰. According to Casciotti et al. (2007), the here measured values are 9‰ lower than expected according to a situation where $\delta^{18}O$ values of nitrite was established in equilibrium with lake water with a $\delta^{18}O$ of -10‰. However, Sebilio et al. (2019) found that the $\delta^{18}O$ values of nitrite are lower (< +4‰), as observed in our study, when the oxygen exchange reaction is controlled by biotic exchange processes compared to those suggested by Casciotti et al. (2007), who studied abiotic exchange reactions between nitrite and water-oxygen.

**Reviewer #2**

We like to express our gratitude for the detailed feedback from Reviewer #2. Below, we have provided a detailed point-by-point list of answers and replies to the comments and suggestions raised by the reviewers. We have had every attempt to address all suggestions and the numerous highly valuable recommendations where appropriate, and have provided detailed responses and explanations below.

Response to general comments:

Reviewer #1 mentioned that the authors seem well aware of the limitations of their isotope results and temper their conclusions with an appropriate acknowledgement of the limitations of the presented data (with very few exceptions where a slight over-reach of data interpretation can be identified). In contrast, Reviewer #2 asked for a more cautious interpretation of the data.

Therefore, we have settled in the revised manuscript on a compromise approach.

To address this comment, we acknowledge in the revised version of the manuscript the limitation of isotope and microbial community data when activity indicators, for example derived from NanoSIMS analyses (L 419), are missing (for example: Ls 330, 362, 426, ..).

We also agree with reviewer #2 that microcosm and incubation experiments are excellent approaches to evaluate which processes govern isotope profiles similar to those observed in our study or to determine degradation rates. We are in fact pursuing such experiments (e.g. Kuloyo, 2020). However, it is also known that isotopic fractionation factors especially for transformation processes in the nitrogen cycle observed in the laboratory are not always transferrable to field sites (e.g. Brunner et al. 2013), and hence are a useful complement but not a replacement of field studies. Similarly, incubation experiments would also show a potential for processes (rates) that could be occurring *in situ*.

Reviewer #2 stated that the paper is prepared with a "certain degree of carelessness with a lot of typos and word adding", but marked only a few typos within the manuscript. We have addressed all typos and stylistic improvements recommended by Reviewer #1 and #2 and tried to eliminate any remaining spelling and grammatical mistakes, in order to address the concerns of Reviewer #2.

Reviewer #2 expressed some concerns with the modelling portion of the manuscript. The modelling component in the original manuscript constituted only a minor part of our study and was rather used to develop the hypothesis. We agree that the application of a rather simple model demands a more cautious interpretation of the modelling results (micro-aerobic methane oxidation) and we have modified the revised manuscript accordingly. For instance, the conclusion that micro-aerobic methane oxidation will only occurring to a very limited extent, must be verified with a numerical model including concentration and isotope profiles that will be published elsewhere (see point-by-point answers)

In this context Reviewer #2 asked whether "steady-state conditions" can be assumed and mentioned that our modelled concentrations will be way off by using Kz of 0.1 to 2.1 $m^2$/d. We now explicitly state in the "Method" part of the revised manuscript that the studied ecosystem is a hydraulically static system, where we assume that advection or mixing do not occur. However, diffusion has to be modelled dynamically, in order to reflect system dynamics adequately and "steady-state conditions" cannot be assumed. Now it is stated in the revised manuscript (§2.4) (L119), that we assumed static conditions that means that there is no mixing and advection. These conditions can be assumed for lake Fohnsee between June to September and were therefore valid for our sampling campaign.

Reviewer #2 also commented on the use of Kz from literature data. This issue is addressed in detail in the point-by-point responses. In short, the suggested Kz value by the reviewer of 0.004 $cm^2$/s (0.03 $m^2$/d) is a factor of ten too small compared to (typical) literature data (e.g. Oswald, 2015 and others). If we calculate the Kz that is valid for the investigated lake (as suggested by Rev. #2) than the new $K_z$ value fits perfectly with literature data (0.1 $m^2$/d), and the value that was used in the original draft of our manuscript (see also point-by point-answer). As also observed by Reviewer #2, we agree that field data and modelling results do not fit, and hence this supports our conclusion that only diffusion cannot describe the depth-profile of methane concentrations. Consequently, we have also calculated the degradation of methane using the fitting parameter k which represents the first order rate constant.

Reviewer #2 also commented that the statement in the abstract "that it is an exaggeration to state that nitrate-dependent methane oxidation has the potential …. is not really convincing", while stating that "it does not happen; we know that". Unfortunately, no references were provided that would conclusively document that this process is not occurring at any study site.

In contrast, there are a few studies demonstrating by genome analysis and anaerobic experiments with enrichment cultures that Crenothrix and other microbes can reduce nitrate with methane to $N_2O$ (Oswald et al. 2017, Mustakhimov et al. 2013, Naqvi et al. 2018 etc). Therefore, it is worthwhile to discuss the occurrence and potential of this process to reduce nitrate loading in aquatic environments. These previous publications suggested that this newly discovered process (AOM with nitrate/ nitrite linked with anammox) could have environmental relevance. In this regard, we believe that our study and new data provide a valuable contribution to the literature on this topic.

Below we provide a point by point response to the comments of reviewer #2 below. Our responses appear in regular font while the original comments by the reviewers appear in italic font.

*Comments of Reviewer #2*

Some of the comments of Reviewer #2 were already discussed at the beginning of our point-by-point answers.

Main points:

*Abstract: In light of much more important N-loss reactions (denitrification anammox) I think it is an exaggeration to state that nitrate-dependent methane oxidation has the potential to reduce nitrate loading. It just does not happen, we know that. I doubt that AOM-denitrification-anammox process really is an overlooked process...it simply is less important than canonical denitrification and aerobic methane oxidation.*

**Response**: (see discussion above)

L 15: The nitrite (n-damo)/ nitrate dependent anaerobic methane oxidation and the anaerobic oxidation of ammonium (anammox) represent two microbially-mediated processes that can reduce nitrogen loading of aquatic ecosystems and associated methane emissions to the atmosphere.

*There is no information on the site/lake. The name and location of the lake should at least be mentioned*

**Response**: In the revised version of the manuscript, the name of the lake has been added to the abstract and we have re-written the first sentence as follows:

L 17: Here, we report vertical concentration and stable isotope profiles of $CH_4$, $NO_3^-$, $NO_2^-$ and $NH_4^+$ in the water column of Lake Fohnsee (Southern Bavaria, Germany) that may indicate linkages between denitrification, anaerobic oxidation of methane (AOM) and anammox.

*L56: Why is nitrate reduction more important in lakes than nitrite reduction, just because there is more nitrate than nitrite? Nitrite is an intermediate and assuming that the most important N*

*loss pathway is complete denitrification, nitrite reduction has to balance nitrate reduction, if nitrite does not accumulate.*

**Response**: We have deleted this sentence

*L88: "...coupled the diffusion model with a degradation term to clarify the effect of dissolved oxygen on methane oxidation. The observed coupled process has the potential to constitute an important sink of dissolved nitrogen (NO3-, NO2-, NH4+) and methane(CH4) in freshwater environments." What exactly is coupled? What coupled process are the authors referring to? This is not clear at this point of the article what they did in the model and how O2 thresholds are integrated. Even if an explanation will follow in the method section, this needs to be clarified (or moved to the more detailed sectionson the model parametrization).*

*Model: There is not enough explanation of the model. Obviously, it is not a real reaction-diffusion model, but it also is not just a diffusion model, right? What are there action parameters, how are they set? I am not an expert in modelling, but it remains unclear how the modelling works, apparently, a purely diffusive part and a reaction partis combined, but the coupling of the model components is unclear. Most importantly ,how well constrained is turbulent diffusion? The results (modelled concentration pro-files and isotope ratios in water column) will be highly sensitive to the choice of the D,and adopting values for D (by the way D is used usually for molecular diffusion only) from other lakes may not be appropriate. In fact, the authors seem to have a very limited knowledge of modelling turbulent diffusion in lakes. Firstly, it seems that their choice of what they call D (or K in the literature) is at least two orders of magnitude higher than would be expected for a stratified lake. They cite Oswald et al. from which D was adopted. But looking into the paper by Oswald, I saw that their choice of Kz was 4x10-3 cm2 s-1, which corresponds to approx. 0.035 m2 d-1. If the authors really used D/Kz values between 0.1 and 2.1, their modelled concentrations will be way off. Finally, assuming different turbulent diffusion coefficients for O2 and CH4 is non-sense. Turbulent diffusion is not solute-specific (in contrast to molecular diffusion), itis a hydrodynamic property of the flow field. As for the first-order methane oxidation rate coefficient, how can the authors just assume a value adopted from other studies? This parameter will change significantly between ecosystems, and has to be estimated based on fitting of the model to the observational data.*

**Response**: We have re-written this part of the Methods section and have already given our view to this comment at the very beginning of the point-by-point answers.

In addition, we have written Kz instead of D, have calculated $K_z$ for Lake Fohnsee and have clarified that the newly calculated Kz fits with literature data. We have already mentioned above that the reviewer's data from the literature seem not appropriate for this case study.

In the revised text we now use the 1D diffusion model with a degradation term to find some evidence that the observed concentration profile of methane cannot only be explained by

diffusion. Subsequently, we have fitted the theoretical methane concentration depth-profile to the field data using the estimated Kz value for the lake and the rate constant k as fitting parameters and compared the fitted k-value with literature data. The k-value is a characteristic parameter for sites with similar environmental conditions and may give a first hint of AOM with denitrification but does not provide compelling evidence, as outlined in our revised manuscript (from L 119; from L 295)

Furthermore, there is no difference in using a k-value as fitting parameter, as mentioned by the Reviewer, or using literature data for k that are characteristic for the studied redox process and evaluate the quality of fitting.

New:

From lines 299

*Nitrate/nitrite isotope measurements: The authors write: Nitrogen and oxygen isotope ratios of nitrate were calculated by measuring nitrite alone as well as the mixture of nitrite and nitrate in a sample and using an inverse mixing calculation to determine the isotopic ratios of nitrate alone. First of all, there seems to be a duplication in this sentence. I think I understand what the authors did. They measured the isotopic composition of nitrite, and then the isotopic composition of the mixture. Based on mass balance calculation, they then calculate the isotopic ratios of nitrate alone. This works for d15N, but does it work for d18O? I am pretty sure that it does not. In a sample that contains nitrite and nitrate, O isotope fractionation during the conversion to N2O is different for nitrite and nitrate. Hence the d18O of the N2O cannot simply be standardized, because the O-isotope offsets will be different for nitrite and nitrate. In other words, the d18O of a NOx sample is probably meaningless, and so will be the calculated d18O nitrate values. The nitrate d18O should have been measured after removal of the nitrite. Could the changes in Dd15N(nitrate-nitrite) be an artifact that is simply the result of this effect and changing nitrite/nitrate concentration ratios?*

**Reponse:** (from L164)
Regardless of the method used (bacterial reduction, reduction with cadmium or most recently with titanium), the objective is the production of $N_2O$. The method used for this study was the reduction of nitrate to nitrite on an activated cadmium column and then the conversion of nitrite to $N_2O$ with an azide buffer. This method has the advantage of being able to test the conversion yields at each stage. For each step, international standards are used. In very rare cases in the environment, a significant amount of nitrate and nitrite may be present. Our approach is based not on the addition of an additional reagent, which could also create a bias, but on the conversion of nitrite to $N_2O$ and the conversion of the nitrate+nitrite mixture to $N_2O$. Details of the calculations were recently published by Sebilo et al. (2019) in Scientific Reports and we now refer to this publication.
The calculation of the isotopic composition is based on the measurement of the isotopic composition of $N_2O$ with an IRMS and the correction between the values obtained for the standards and the values measured for unknowns (e.g. samples) by linear regression. For

samples obtained from 14, 16, 18 and 20m depth, both nitrite and nitrate were present. However, taking into account the concentration ratios, the amount of nitrite represents at most 10% of the total concentration for the samples except for the 20m sample where the nitrite concentration is around 1 mg/L and the nitrate concentration is around 0.5 mg/L. For this depth, taking into account the two molecules and calculating the nitrate $\delta^{18}O$ gives a value of 5.6‰ whereas it was 5.4‰ without correction.

*L188: How was complete outgassing of CH$_4$ assured before headspace analysis? Was brine/NaOH added? Concentrations were calculated based on Henry's Law, but what about the d13C? Is there an isotope shift between CH4 in the headspace and the CH4 dissolved? If so, was that considered?*

**Response**: (from L 190)

- outgassing was not complete since, since we followed the headspace equilibration technique by EPA (2002);

- following this EPA technique, we did not add either a salt solution or NaOH to the sample solution;

- We assumed negligible C isotope fractionation between dissolved methane and methane in the headspace (e.g. Feux 1980) and therefore report the measured $\delta^{13}C$ values for headspace methane.

*Results: I am a bit disappointed by the low number of data points/analyses. As a consequence, isotope gradients are not well resolved (and their interpretation is hence complicated), and the profiles are not replicated for several time points. Do the authors assume steady state conditions? How relevant is this for the model fitting?*

**Response:** We agree that a depth resolution of 0.5 m or even lower would have been better, but we sampled up to 2L of lake water for each depth (water samples for IC, isotopes, DOC and microbiology) and we wanted to exclude mixing of water from different depths by sampling.

*Figure 3c is very difficult to read? Why not showing profiles (connected symbols) for the most relevant OTUs. It is almost impossible to see the vertical structure.*

**Response:** We now present Figure 3c with the symbols connected for the most relevant OTUs to make it easier to see the vertical structure.

*Discussion: It is not clear to me what the arguments are that allow the authors to exclude oxic methane oxidation. I agree that the concentration profile suggests reaction below the redoxcline, but you do not need to model this to come to this conclusion. At the same time, do the authors assume steady state? Apparently, the lake undergoes seasonal fluctuations, so that*

*the curvature of the concentration profiles may represent a non-steady state, and its interpretation with regards to where reaction takes place and where not is biased.*

**Response**: We agree that a more complex model is needed to perform flux estimation and to evaluate if micro-aerobic methane oxidation has occurred near the oxycline. In addition, it will make sense as suggested by Reviewer #2 to incorporate the isotope compositions of methane in the modelling part, but this was out of scope for this paper and the former manuscript (L 303).

We have improved the modelling part and have shown the effect of diffusion and degradation on the depth-concentration profile of methane (L 295) and have rewritten this part of the manuscript.

L 303: However, because the oxic-anoxic transition zone is in close proximity to the nitrate reduction zone, numerical modelling studies are required that link the stable isotope ratio and concentration profiles of methane to study the effect of micro-aerobic methane oxidation near the oxycline at lake Fohnsee. (Fig. 4).

*Can the authors explain why a 90% decrease in ammonium is associated with a 15N shift of only 4‰*

*L335-7: The authors say that above 20 m water depth, there is no evidence for ammonium oxidation. Why? Because the d15NH4 values do not increase? But they also do not increase much below that depth, where the authors suggest that anammox occurs. And most strikingly, the ammonium profile is essentially linear all the way up to the oxycline. To me this suggests that not much ammonium oxidation is taking place at this depth, and essential all NH4 is oxidized at the oxycline.*

**Response**: We do not understand that Reviewer #2 mentioned a 90% decrease in ammonium concentration that is linked to a small N isotope effect in the remaining ammonium. We have observed a decrease of ammonium concentration from 1 mg/L at the bottom of the lake to 0.8 mg/L at a depth of 20 m and simultaneously an increase of $\delta^{15}N$ of ammonium of 4 ‰. Above this depth $\delta^{15}N$ values of ammonium remain constant. Above a depth of 20 m, ammonium concentrations decrease from 0.06 mmol/L to below the detection limit at a water depth of 12 m, probably by diffusion where no significant isotope fractionation is expected and micro-aerobic $NH_4$-oxidation.

To explain the small nitrogen isotope fractionation effects, Wunderlich et al. (GCA 2018) have suggested a transport limitation model, that is outlined in lines 345 to 353.

*340-345: The authors cite the anammox isotope effect study by Brunner and colleagues. But they mix up equilibrium and kinetic N isotope effects between nitrite and nitrate. The inverse kinetic N isotope effect, which applies to active nitrate production from nitrite by anammox, is much lower than the -61‰ mentioned.*

**Response:** Right, the equilibrium N isotope effect between nitrate and nitrite is −60.5 ± 1.0‰ whereas the inverse kinetic N isotope fractionation associated with the oxidation of nitrite to nitrate is −31.1 ± 3.9‰. The latter fits very well to our field data The explanation for small nitrogen isotope effects during anammox is also given in L 345-355.

*The authors should explain better why anammox could produce a d18O vs d15N NO3 relationship of 0.5. Is this slope consistent with nitrate production from nitrite with the incorporation of O atoms from water? Such slopes in d18O vs d15N NO3 plots have been observed in several ground/freshwater studies. Does this imply that in all these environments anammox was the main N-loss pathway?*

**Response:** Indeed, slopes lower than 1 can also be observed at aerobic/ anaerobic interfaces such as groundwater systems, the oxycline of stratified lakes or as outlined in Wunderlich et al. (2018 in GCA) by specific organisms that re-oxidize nitrite to nitrate and do not imply that anammox is the main N-loss pathway in all environments. Please note that we consider a system that may be controlled by anaerobic redox conditions. This was also clearly stated in the manuscript to demonstrate that in a $\delta^{18}O$ vs. $\delta^{15}N$ plot for nitrate a slope lower than 1 is a powerful indicator for the occurrence of anammox in an anoxic environment.. Furthermore, we developed several lines of evidence to come to the conclusion that anammox may have occurred.

From L 335

The comment whether the slope is consistent with nitrate production from nitrite with the incorporation of O atoms is somehow exaggerated because we think that the reviewer knows that this depends on the enzymes involved etc. Here we relegate to the paper of Granger and Wankel (2016) in PNAS.

*What is the relative abundance of "normal" nitrate and nitrite reducers compared to NC10 and Crenothrix?*

**Response**: Because nitrate and nitrite reduction is such a widespread trait held by many facultative anaerobic bacteria, it is not possible to use our 16S rRNA gene sequence data to specifically show the abundance of 'normal nitrate and nitrite reducers' as the reviewer requests. However, the Gamma proteobacteria are very abundant in our samples, and are well known to have many species that are capable of nitrate and nitrite reduction, a trait that is widespread throughout this class. Since the Gamma proteobacteria relative abundance increases with depth into the anoxic zone (Fig 3b), it is likely that many of the Gamma proteobacteria in deeper waters of the lake are responsible for nitrate and nitrite reduction, and denitrification. We will add an explanation to the revised manuscript (from line 426).

Some minor points:

These few suggestions were accepted and improvements were made in the revised manuscript.

*I am wondering about the fact that there was not an appropriate d18O_NO2 standard used.*

**Response**: We agree and have deleted the $\delta^{18}$O values of nitrite from the revised manuscript following the suggestion of both reviewers.

*I am still not convinced by the way they sell their data set (evidence for true AOM is weak), and I still think there are some issues, but I guess that this is part of the scientific exchange. We do not need to all agree. It is definitely a nice data set, certainly worth to be published.*

**Response**: We agree with Reviewer #2 that a more refined depth resolution of the samples and the subsequently obtained data would have been beneficial for further strengthening the evidence for AOM; as pointed our previously in the manuscript, this was, however, not possible and, therefore, we have very carefully worded this section by stating "some evidence …." in lines 294, 326, and 333

*The authors tone down in their response the value of incubation experiments. I do not agree. Experiments, even if not directly reflecting the natural conditions, would have helped to calibrate the observed isotopic signatures, and to gain confidence in their interpretation. I was not referring to isotope effect experiments, and I am aware that rate measurements will deliver potential rates only, but clearly, rate measurements would have helped to indicate the potential for the use of specific substrates, and thus would have helped to support (or not) the claimed links between different biogochemical reactions (anammox and AOM).*

**Response**: We thank the reviewer for this clarification. Fact is that we have not conducted any complementary laboratory experiments but we agree that laboratory experiments could help to study the potential for the use of specific substrates.

*The authors state that there is barely any mixing during summer ("no mixing, no advection"); I assume they mean no advective mixing, because turbulent diffusive mixing, even if sluggish will take place.*

**Response**: This is an excellent point and we have modified the manuscript accordingly. Now line 134 reads as follows: This corresponds to the period where stagnant conditions for lake water are assumed to prevail (no advective mixing)…..

As for the choice of a value for Kz: In my first review; I was not suggesting a value of 0.03 m2/d. I was referring to a paper by Oswald et al. from Lake La Cruz (Oswald et al. 2016), for which this value was reported. I mixed up the different studies/lakes investigated by Oswald et al., and values from Lake Rotsee, as well as values for Lake Lugano seem to be higher, indeed. But the broad range of Kz values observed for the different meromictic lakes highlights that there is not "a typical literature value", and that just assuming a value for Kz is difficult.

**Response**: Thank you for this clarification.

*As for the interpretation of the NH4 concentration versus the d15NH4 profiles: I see a more or less steady decrease from 1 to <0.1 mgN/l between 22 and 12 m, without a strong overall 15N*

*enrichment. From these patterns, I find it quite difficult to pinpoint anaerobic ammonium oxidation in the deep water column. Within the error of the d15NH4 analyses, the isotope profile looks almost straight to me....an observation that could most plausibly be explained by aerobic ammonium explanation at the redox transition zone, which serves as an efficient NH4 sink without much N isotope fractionation.*

**Response**: We clearly stated from line 342 to 344 where ammonium concentrations decreased and $\delta^{15}N_{NH4}$ increased:

The decrease in ammonium concentration with decreasing water depth is accompanied by an enrichment of $^{15}N$ in the remaining ammonium shifting the $\delta^{15}N_{NH4}$ values from 7.9‰ to 11.6‰ between 22 and 20 m water depth (Fig. 2C), suggesting that ammonium is oxidized anaerobically while enriching the remaining substrate in $^{15}N$.

We also like to point out that we did not interpret the N isotope ratios and concentrations of ammonium in isolation, but considered the following observations in combination:

- both nitrite and ammonium concentration trends;
- $\Delta\delta^{15}N$ between nitrate and nitrite increased from 11‰ in NMTZ to > 26‰ at the water depth of 20 m, where $\delta^{15}N$ values of ammonium increased while $NH_4^+$ concentrations decreased (Fig. 2C);
- the deviation of the slope of $\delta^{18}O$ versus $\delta^{15}N$ values on a dual isotope plot (2D plot) for nitrate from the expected value of 1 for microbial denitrification is a powerful tool to identify anammox.
- This, together with the presence of '*Candidatus Anammoximicrobium*' within the strictly anaerobic water column, is in our view sufficient to come to the conclusions that these combined observations may indicate that anammox has occurred at Lake Fohnsee between a water depth of 20-21 m.

Reviewer 1 also expressed the opinion that the statements about the key findings are reasonably tempered in the revised manuscript.

*I am still confused with regards to the d18O_NOx values. Again, in a sample that contains nitrite and nitrate, O isotope fractionation during the conversion to N2O (independent of the method of conversion) must be different for nitrite and nitrate (via nitrite), simply because one versus two O atoms will be plucked off during the fractionating transformation, respectively. Hence it remains unclear to me how the d18O of the combined NO3+NO2 sample is standardized. The problem is not the yield, the problem is that the N2O from the nitrate will likely have a much higher d18O than the d18O of the N2O generated from the NO2.*

**Response**: As mentioned in our response to Reviewer #1, we agree with the major limitation of the $\delta^{18}O$ values of nitrite as no reference material was used, which also impacts mixing calculations. However, for most of the nitrate samples (14m to18m water depth) nitrite represents at most 15% of the total nitrate+nitrite concentration. Recalculations using a value of +4‰ for $\delta^{18}O$ of nitrite revealed that the $\delta^{18}O$ values of nitrate are between 2 and 3‰ lower compared to those calculated with the previously reported $\delta^{18}O$ values of nitrite (ca. -4 to +1‰). These changes had no consequences for our conclusions reported in the earlier version of the manuscript.

In the revised manuscript we have re-calculated the $\delta^{18}O$ values of nitrate.

Now from line 175 the revised manuscript reads: For $\delta^{18}O$ values of nitrate we performed a mass-weighted isotope mass balance calculation assuming that at a pH of 7 the $\delta^{18}O$ of nitrite is in equilibrium with water with a value close to +4‰.

Figs 2C and 5 were corrected using the new data set of $\delta^{18}O$ of nitrate and additional information was given for Fig. 2C.

*L915: established is misspelled*

**Response:** This part was deleted

In summary, we have made every attempt to address the suggestions and the highly valuable recommendations by both reviewers, and have provided detailed responses and explanations above. We hope that the revised manuscript can now be accepted for publication.

References:

KULOYO, O., RUFF, S. E., CAHILL, A., CONNORS, L., ZORZ, J. K., HRABE DE ANGELIS, I., NIGHTINGALE, M., MAYER, B. & STROUS, M. (2020): Methane oxidation and methylotroph population dynamics in groundwater mesocosms. Environmental Microbiology, 22(4): 1222-1237 (early access February 2020; published April 2020). https://doi.org:10.1111/1462-2920.14929

SEBILO, M., ALOISI, G., LAVERMAN, A. M., MAYER, B., PERRIN, E., VAURY, V., MOTHET, A. & LAVERMAN, A. M. (2019): Controls on the isotopic composition of nitrite ($\delta^{15}N$ and $\delta^{18}O$) during denitrification in freshwater sediments. - Scientific Reports, 9: 19206. Doi.org/10.1038/s41598-019-54014-3; published December 16, 2019.

WUNDERLICH, A., HEIPIEPER, H.J., ELSNER, M. AND EINSIEDL, F. (2018) Solvent stress-induced changes in membrane fatty acid composition of denitrifying bacteria reduce the extent of nitrogen stable isotope fractionation during denitrification. Geochimica et Cosmochimica Acta 239, 275-283.